# Healthcare waste management practices and associated factors among healthcare workers in Sub-Saharan Africa: A systematic review and meta-analysis

Gete Berihun[1]*, Zebader Walle[2], Belay Desye[3], Chala Daba[3], Abebe Kassa Geto[4], Lake Kumlachew[1], Leykun Berhanu[3]

**1** Department of Environmental Health, College of Medicine and Health Sciences, Debre Markos University, Debre Markos, Ethiopia, **2** Department of Public Health, College of Health Sciences, Debre Tabor University, Debre Tabor, Ethiopia, **3** Department of Environmental Health, College of Medicine and Health Sciences, Wollo University, Dessie, Ethiopia, **4** Department of Public Health, College of Medicine and Health Sciences, Woldia University, Woldia, Ethiopia

* geteberihun@gmail.com, gete_berihun@dmu.edu.et

## Abstract

### Introduction

Inadequate management of healthcare waste present significant health hazards to healthcare workers, patients, waste handlers, and the whole communities, especially in developing countries. Although various primary studies have been conducted in different countries across the continent, there has been no comprehensive research examining healthcare waste management practices in Sub-Saharan Africa.

### Objective

This review aimed to assess healthcare waste management practices and associated factors among healthcare workers in Sub-Saharan Africa.

### Methods and materials

This systematic review and meta-analysis were performed using the Preferred Reporting Items for Systematic Reviews and Meta-Analysis (PRISMA 20) guidelines. PubMed, Science-Direct, Google Scholar, Hinari, and Google databases were used to find essential literature. The extracted data were analyzed using statistical software, STATA version 14. Publication bias was assessed using the Egger test and funnel plot, whereas heterogeneity was assessed using the $I^2$ statistic.

### Results

This review include 29 studies comprising 7588 participants. The pooled estimate of good healthcare waste management practices among participants was 49.74%

**Data availability statement:** All relevant data are within the paper and its Supporting Information files.

**Funding:** The author(s) received no specific funding for this work.

**Competing interests:** The authors have declared that no competing interests exist.

(95% CI: 43.73–55.76) ($I^2 = 96.8\%$, $P < 0.000$). Sex, knowledge, training on healthcare waste management, use of working manuals/guidelines, and working hours were factors significantly associated with healthcare waste management practices among healthcare workers., Studies done in South Africa reported the highest good healthcare waste management practices with a value of 54.34% (95% CI: 48.05, 60.63), $I^2 = 0.00\%$, $P < 0.00$. The pooled estimate of good healthcare waste management practices before the occurrences of COVID-19 pandemic was 50.49% (95% CI: 40.7, 60.25), ($I^2 = 97.9\%$, $P < 0.000$). Public health facilities also reported having lower waste management practices with a value of 46.86% (95%CI: 39.33, 54.38%), $I^2 = 96.8\%$, $P < 0.000$.

## Conclusions

This review showed that only half of the healthcare workers practiced good healthcare waste management practices. Sex of the healthcare workers, training status, use of working manuals/guidelines, knowledge towards healthcare waste management, and their daily working hours were factors significantly associated with healthcare waste management practices among healthcare workers. Hence, respective healthcare authorities should develop and implement different healthcare waste management strategies, including ongoing in-service training, provision of healthcare waste management manuals, and conducting regular monitoring to enhance healthcare workers' knowledge and practices towards healthcare waste management practices.

## Introduction

Biomedical waste refers to any solid or liquid waste generated during the diagnosis, medication, or immunization of humans or animals, as well as wastes generated from related research activities, or the production and testing of biological products, and is contaminated with human fluids [1–4]. The UN Basel Convention considered it as the second most harmful type of waste following nuclear waste [5]. Healthcare waste management (HCWM) encompasses waste generation, segregation, transportation, storage, treatment, and disposal practices [4,6]. The advancement in medical care and the introduction of more advanced equipment have made dramatically increased the amount of waste generated per patient in healthcare facilities across the globe [7]. Literatures revealed that almost 85% of the healthcare waste is general waste. The remaining 15% of the total waste produced is hazardous wastes comprising of 10% infectious and 5% non-infectious. These types of wastes typically contains sharps, genotoxic waste, heavy metals, chemicals, and pharmaceuticals that can be contagious, toxic, or radioactive, requiring urgent and careful attention from the international communities [4,8,9].

In Africa, there are an estimated of 67,740 health facilities which produce an approximately of 282,447 tons of medical waste every year [10]. In Sub-Saharan

Africa, the segregation of medical waste according to its characteristics, the use of recommended color-coded containers for waste collection, and the storage in an isolated area are generally inadequate and do not meet satisfactory standards [11–14]. Additionally, healthcare workers do not utilize personal protective equipment and accessories in their working healthcare facilities [11]. Furthermore, the waste is usually dumped either into their backyard in a simple pit or into open garbage bins on the roads, which causes further damage to public health and the environment at large.

Poor healthcare waste management jeopardizes the health workers, employees who handle healthcare waste, waste pickers, patients and their families, and the community at large to infection, toxic effects, injuries, and risks of polluting the environment [15–19]. Each year, improper healthcare waste management practices leads to an estimated 5.2 million deaths globally, including 4 million among children [20]. Additionally, more than 2 million healthcare workers are infected with over 30 blood-borne pathogens [21]. Exposure to hazardous waste containing mercury and expired drugs, particularly cytotoxic, cytostatic, and antibiotic drugs, causes serious health problems, including cancer, mutations, teratogenicity, respiratory system, and skin diseases [22]. Nosocomial infections, including Hepatitis B and Hepatitis C cases, and 2–3% of HIV infections prevalent as a result of poor waste management practices [23–27]. The growth of multidrug-resistant microbes originating from healthcare waste is another major challenge facing the global community [28].

The poor healthcare waste management practices may be attributed because of poor attitude and lack of knowledge on proper waste management, absence of policies or regulations, financial scarcity, and lack of waste management equipment. These problems are highly prevalent in low- and middle-income countries, where healthcare facilities are rapidly expanding but technological and financial constraints may hinder its management [15,17]. Healthcare facilities in these regions do not even adhere to policies, and no effective national health authorities to oversee and regulate HCWM [8,9,29–31]. It was hard to find a comprehensive study focusing on healthcare waste management (HCWM) practices among healthcare workers in SSA. Therefore, this study aimed to provide a comprehensive overview of HCWM practices by compiling and summarizing existing primary studies on the prevalence and factors associated with healthcare waste management practices in SSA.

## Methods and materials

### Protocol and registration

The protocol for this systematic review was formally registered with the International Prospective Register of Systematic Reviews (PROSPERO), hosted by the University of York Centre for Reviews and Dissemination. The review protocol was registered on December 22, 2023, with the assigned record code of CRD42023492372.

### Information search and search strategies

This systematic review and meta-analysis were conducted using the Preferred Reporting Items for Systematic Reviews and Meta-Analyses (PRISMA-2020) guidelines [32]. (S1 File) To identify the relevant articles for inclusion, a comprehensive search of literature was conducted using multiple databases, including PubMed/MEDLINE, Science-Direct, Hinari, and Google Scholar. Additionally, Google search was used to search out additional articles which are not indexed in the above mentioned databases. Furthermore, reviewing the reference lists of relevant studies and consulting with content experts to identify any grey literature (unpublished studies or reports) was carried out accordingly. For the PubMed/MEDLINE database search, a combination of key terms and Boolean operators (AND, OR) was used to construct the search strategy. ("Healthcare waste" [all field] OR "medical waste" [all field] OR "Biomedical waste" [all field] OR "clinical waste" [all field] OR "Hospital waste" [all field] "Infectious waste" [all field] OR "Healthcare –associated waste" [all field] OR "Patient care waste" [all field]) AND (" associated factors" [all field] OR "contributing factors" [all fields] OR " determining factors" [all fields] OR "influencing factors" [all fields]) AND ("Angola" [all fields] OR "Benin" [all fields] OR "Botswana" [all fields] OR "Burkina Faso" [all fields] OR "Burundi" [all fields] OR "Cabo Verde" [all fields] OR "Cameroon" [all fields] OR "Central African Republic"

[all fields] OR "Chad" [all fields] OR "Comoros" [all fields] OR "Democratic Republic of the Congo" [all fields] OR "Djibouti" [all fields] OR "Equatorial Guinea" [all fields] OR "Eritrea" [all fields] OR "Eswatini" [all fields] OR "Ethiopia" [all fields] OR "Gabon" [all fields] OR "Gambia" [all fields] OR "Ghana" [all fields] OR "Guinea" [all fields] OR "Guinea-Bissau" [all fields] OR "Ivory Coast" [all fields] OR "Kenya" [all fields] OR "Lesotho" [all fields] OR "Liberia" [all fields] OR "Madagascar" [all fields] OR "Malawi" [all fields] OR "Mali" [all fields] OR "Mauritania" [all fields] OR "Mauritius" [all fields] OR "Mozambique" [all fields] OR "Namibia" [all fields] OR "Niger" [all fields] OR "Nigeria" [all fields] OR "Republic of the Congo" [all fields] OR "Rwanda" [all fields] OR "São Tomé and Príncipe" [all fields] OR "Senegal" [all fields] OR "Seychelles" [all fields] OR "Sierra Leone" [all fields] OR "Somalia" [all fields] OR "South Africa" [all fields] OR "South Sudan" [all fields] OR "Sudan" [all fields] OR "Tanzania" [all fields] OR "Togo" [all fields] OR "Uganda" [all fields] OR "Zimbabwe" [all fields]).

**Eligibility criteria. Inclusion criteria:** Articles that fulfilled the following criteria were included in this review.

**Population**: The study participants for this review must be healthcare workers employed in healthcare facilities in SSA.

**Outcome variables**: The articles must have quantitative data on the proportion of HCWM practices (good/ poor) and factors associated with management practices.

**Study Design**: The review includes a cross-sectional study design.

**Study Setting**: The study setting is SSA.

**Language**: Only full-text articles published in English were included in the study.

**Publication Period**: Articles that were published between 2018–2023

**Exclusion criteria.** This review excluded qualitative studies, systematic reviews, letters to the editor, short communications, and commentaries. Additionally, articles that could not be fully accessed despite three attempts to reach out to the corresponding author were also excluded from the review.

**Study selection.** GB and CD independently evaluated articles identified through the comprehensive search by screening their titles, abstracts, and full texts to identify articles that met the predefined eligibility criteria. The articles deemed eligible by both independent reviewers were combined. Any disagreements between the two independent reviewers were resolved through evidence-based discussion. In cases where the two reviewers could not reach an agreement, a third independent reviewer, BD, was consulted to make the final decision on whether to include or exclude the disputed article.

**Data extraction and management.** A standardized data extraction form was utilized for the eligible studies, collecting pertinent variables such as the author's name, year of publication, country, data collection method, sampling technique, sample size, proportion of HCWM practices, and assessment of the risk of bias. EndNote reference manager software was employed to manage the search results and eliminate duplicate articles.

**Quality assessment of studies.** The Joanna Briggs Institute (JBI) quality assessment tools for analytical cross-sectional studies were used to assess the quality of the included studies using the following indicators, with the response options of yes, no, unclear, and not applicable. (1) inclusion and exclusion criteria; (2) description of the study subject and study setting; (3) use of a valid and reliable method to measure the exposure; (4) standard criteria used for measurement of the condition; (5) identification of confounding factors; (6) development of strategies to deal with confounding factors; (7) use of a valid and reliable method to measure the outcomes; and (8) use of appropriate statistical analysis. (S1 File) The risks for biases were classified as low bias (a score of 6–8), moderate bias (3–5), and high bias (0–2). Hence, articles with low and moderate biases were included in the review [33]. (S2 File)

## Outcome of interest

This review has two outcome variables. The primary outcome variable is the pooled estimate of HCWM practices, which was measured as either good/ poor practices, presented in terms of percentage. The second outcome variable of this review was factors significantly associated with HCWM practices among healthcare workers, which was expressed in terms of odds ratio (OR) with a 95% confidence interval (CI).

## Statistical methods and analysis

The data was extracted using a Microsoft Excel spreadsheet and exported to STATA version 14 for further analysis. The heterogeneity among studies included in this review was measured using the $I^2$ statistic, with cutoff points 25–50%, 50–75%, and >75% showing low, moderate, and high heterogeneity, respectively [34]. The overall pooled estimate of HCWM practices among healthcare workers was determined using the meta-prop command in STATA. Subgroup analyses was performed using the variables such as types of healthcare facilities, ownership of healthcare facilities, country, geographic region, study year, sample size, and publication status. Sensitivity analysis was done to assess the influence of each study on the overall pooled estimate. Publication bias was determined using the funnel plot test and Egger's regression test, with a p-value <0.05 [35]. A p-value <0.05 was used to declare the association as statistically significant at a 95% Confidence level. The results were presented using graphs, tables, texts, and a forest plot accordingly.

## Results

### Searching process

The initial comprehensive search across various databases and manual sources yielded a total of 3,321 studies. After removing 784 duplicates, 2,537 studies were screened based on title and abstract. Among these, 2,476 articles were excluded for not meeting the inclusion criteria. The full text of the remaining 61 articles was assessed for full-text eligibility. Following this full-text assessment, 32 studies were excluded (28 articles did not report the outcome of interest, and 4 studies were excluded due to their low quality). Finally, 29 studies were included in this review (Fig 1).

### Characteristics of the included study

All studies included in this review were cross-sectional studies, with a total sample size of 7,588, ranging from 55 to 431 per study. Most of these studies employed structured questionnaires and an observational checklist for data collection. Out of these 29 studies included in this review, 16 studies were from Ethiopia [1,2,8,9,12,18,31,36–44], two studies from Rwanda [45,46], three studies each from Uganda [47–49], and Nigeria [50–52], one study each from Cameroon [53], and South Africa [19], Kenya [54], Tanzania [55], and Zambia [56] (Table 1**)**.

**Pooled estimate of HCWM practices among healthcare workers.** The overall proportion of good HCWM practices among healthcare workers is shown in a forest plot using a random effects statistical model. The pooled proportion of good HCWM practices among the study participants was 49.74% (95% CI: 43.73–55.76; $I^2 = 96.8\%$, P < 0.000) (Fig 2**).**

**Subgroup analysis.** The sub-group analysis was performed using the variables such as study country, geographic region, study year (before and after the COVID-19 pandemic), ownership of the healthcare facilities (private, public, or both), types of healthcare facilities, sample size, and publication status.

The subgroup analysis by country revealed that the pooled proportion of healthcare workers having good practices of HCWM in Ethiopia was 52.42 (95%CI: 44.77,60.08) $I^2 = 98.8\%$, P < 0.000; Rwanda, 50.75% (95% CI: 16.94,84.56), $I^2 = 98.2\%$, P < 0.000; Uganda, 44.01% (95% CI: 6.15,84.87%), $I^2 = 99.2\%$, P < 0.000; Nigeria, 50.98 (95% CI: 39.98, 61.97), $I^2 = 89.1\%$, P < 0.000; Cameroon, 50.00 (95% CI: 40.20, 59.80), $I^2 = 0.0\%$, P < 0.000; South Africa, 54.36 (48.07, 60.65), $I^2 = 0.0\%$, P < 0.000; Tanzania, 36.40% (95% CI: 28.85, 43.95), $I^2 = 0.0\%$, P < 0.000; Zambia 43.10% (95% CI: 38.21, 47.99), $I^2 = 0.0\%$, P < 0.000; and Kenya, 33.0% (95% CI:26.84,39.16), $I^2 = 0.0\%$, P < 0.000 (Fig 3).

Regarding the geographic region, the proportion of good healthcare waste management in east Africa was 49.66% (95% CI: 42.28, 57.05%), $I^2 = 97.4\%$, P < 0.000; West Africa 48.96% (95% CI: 42.27, 55.65%), $I^2 = 81.8\%$, P < 0.000; South Africa 54.34% (95% CI: 48.05, 60.63%), $I^2 = 0.0\%$, P < 0.000 (Fig 4).

Additionally, the subgroup analysis based on the study year indicated that the proportion good healthcare waste management practices among healthcare workers before the occurrence of the COVID-19 pandemic was 50.49% (95%

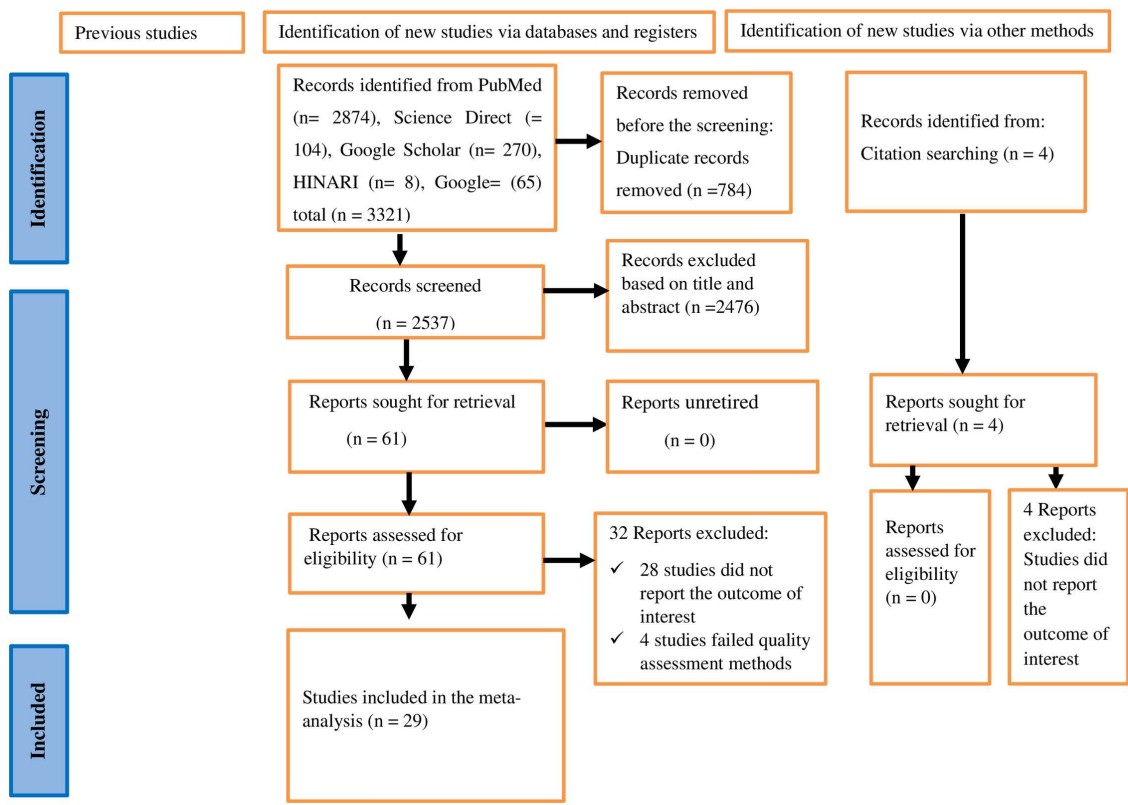

**Fig 1. A PRISMA flow chart showing study selection for systematic review and meta-analysis on the healthcare waste management practices and associated factors among healthcare workers in Sub-Saharan Africa.**

CI: 40.70, 60.25), $I^2 = 97.9\%$, P < 0.000 whereas after the occurrences of the pandemic was 48.83% (95% CI: 42.70, 54.96), $I^2 = 92.7\%$, P < 0.000 (Fig 5).

The subgroup analysis based on the ownership of the healthcare facilities, in public facilities 46.86% (95% CI: 39.33, 54.38%), $I^2 = 96.8\%$, P < 0.000 healthcare care workers demonstrated good healthcare waste management practices whereas in private facilities the proportion of good healthcare waste management practice was 49.25% (95% CI: 41.40, 57.10), $I^2 = 86.7\%$, P < 0.000) (Fig 6).

Furthermore, the subgroup analysis based on the types of healthcare facilities indicated that the proportion of hospital healthcare workers good practices of healthcare waste management practices was 50.24% (95% CI: 43.69, 56.78%), $I^2 = 94.5\%$, P < 0.000; whereas health centers had 71.50% (95% CI: 65.24, 77.76%) $I^2 = 0.0\%$, P < 0.000 (Fig 7).

The subgroup analysis based on publication status, studies which were published had the proportion of good healthcare waste management practices among healthcare workers was 50.45% (95% CI: 44.06, 56.89%), $I^2 = 97.0\%$, P < 0.000 whereas unpublished studies had 40.39% (95% CI: 26.07, 54.72%), $I^2 = 91.9\%$, P < 0.000 (Fig 8).

Finally, studies with a sample size of above the mean had the proportion of good healthcare waste management practices was 48.88% (95% CI: 43.37, 56.39%), $I^2 = 96.8\%$, P < 0.000 whereas less the mean had the proportion of good healthcare waste management practices of 50.64% (95% CI: 40.34, 60.94%), $I^2 = 97.0\%$, P < 0.000 (Fig 9).

**Heterogeneity and publication bias.** The presence of heterogeneity and publication bias was determined in studies included in this systematic review and meta-analysis. Fixed and random effect models are the two main methods used in systematic reviews and meta-analyses. Since this study covers Sub-Saharan Africa,

**Table 1. Descriptive summary of studies included in healthcare waste management practices and associated factors among healthcare workers in SSA.**

| Author, publication year | Study country | Year cat | Sampling technique | Method of data collection | Study design | Sample size | Proportion of good HCWM practice | Risk of bias |
|---|---|---|---|---|---|---|---|---|
| Assemu et.al (2020) [1] | Ethiopia | Before 2020 | SYS | SQ | Cs | 418 | 65.07 | Low |
| Alice et.al (2023) [45] | Rwanda | after 2020 | SRS | SQ | Cs | 200 | 68 | Low |
| Doylo et.al (2019) [18] | Ethiopia | before 2020 | SRS | SQ | Cs | 400 | 42.25 | Low |
| Babirye et.al (2020) [47] | Uganda | before 2020 | SRS | Interview method | Cs | 153 | 10.46 | Low |
| Tolesa (2019) [43] | Ethiopia | before 2020 | SRS | SQ | Cs | 336 | 47.62 | Low |
| Berhanu et.al (2022) [36] | Ethiopia | after 2020 | SRS | SQ | Cs | 411 | 56.2 | Low |
| Wafula et.al (2019) [48] | Uganda | before 2020 | SS | SSQ | Cs | 200 | 71.5 | Low |
| Mariam et.al (2018) [42] | Ethiopia | before 2020 | SRS | SQ | Cs | 320 | 25.31 | Low |
| Gizalew et.al (2021) [8] | Ethiopia | before 2020 | SRS | SQ | Cs | 358 | 29.33 | Low |
| Ibrahim et.al (2023) [38] | Ethiopia | before 2020 | SRS& SS | SQ, OC | Cs | 280 | 56.43 | Low |
| Tilahun et.al (2023) [31] | Ethiopia | after 2020 | SRS& SS | SQ | Cs | 264 | 58.71 | Low |
| Mitiku et.al (2022) [9] | Ethiopia | after 2020 | SRS | SQ & OC | Cs | 431 | 49.42 | Low |
| Sahiledengle (2019) [44] | Ethiopia | before 2020 | SYS | SQ | Cs | 409 | 53.79 | Low |
| Ekanem et.al (2021) [51] | Nigeria | before 2020 | MSS | SQ & OC | Cs | 158 | 58.86 | Low |
| Rutayisire et.al (2019) [46] | Rwanda | before 2020 | SRS | SQ | Cs | 200 | 33.5 | Low |
| Woromogo et.al (2020) [53] | Cameroon | before 2020 | SRS | SQ& interview guide | Cs | 100 | 50 | Moderate |
| Deress et.al (2018) [2] | Ethiopia | before 2020 | Census | SQ & OC | Cs | 296 | 77.36 | Low |
| Olaifa et.al (2018) [19] | South Africa | before 2020 | SS | SQ | Cs | 241 | 54.34 | Low |
| Wassie et.al (2022) [12] | Ethiopia | After 2020 | SRS, SR | SQ, OC | Cs | 278 | 38.8 | Moderate |
| Yakubu et.al (2023) [50] | Nigeria | After 2020 | RS | SSQ, OC | Cs | 371 | 41 | Moderate |
| Millanzi et.al (2023) [55] | Tanzania | After 2020 | RS | SQ | Cs | 156 | 36.4 | Low |
| Leonard et.al (2022) [56] | Zambia | After 2020 | MSS | SSQ, OC | Cs | 394 | 43.1 | Moderate |
| Deress et.al (2019) [40] | Ethiopia | Before 2020 | Purposive | SSQ, OC | Cs | 55 | 80 | Moderate |
| Lemma et.al (2023) [39] | Ethiopia | After 2020 | RS | | Cs | 74 | 64.86 | Low |
| Salaam et.al (2022) [49] | Uganda | After 2020 | RS | SSQ, OC | Cs | 211 | 50.2 | Moderate |
| Thankam et.al (2021) [41] | Ethiopia | After 2020 | RS | SSQ | Cs | 162 | 33.3 | Low |
| Mekassa et.al (2022) [37] | Ethiopia | After 2020 | RS | SSQ, OC | Cs | 254 | 63.4 | Low |
| Mogaka et.al (2023) [54] | Kenya | After 2020 | Census | SSQ | Cs | 224 | 33 | Low |
| Onoh et.al (2019) [52] | Nigeria | Before 2020 | SRS | SSQ | Cs | 234 | 53.9 | Low |

SQ = structured questionnaire; SSQ semi-structured questionnaire; OC Observational checklist; SRS systematic random sampling technique; SS stratified sampling; MSS multi-stage sampling; RS simple random sampling.

where variability in contributing factors is expected, the random effects model is more suitable because of likely substantial heterogeneity across the large geographic area. The fixed effect model fits better when studies are more homogeneous, usually with heterogeneity below 25%. In our case, with heterogeneity of nearly 96.8%, the random effects model is justified and appropriate. The results showed substantial heterogeneity across the included studies ($I^2 = 96.8\%$, p = 0.000). Publication bias was evaluated using both the funnel plot, a subjective method, and the Egger regression test, which provides an objective assessment. The funnel plot appeared reasonably symmetrical, suggesting minimal bias (Fig 10).

The result of Egger's regression test yielded a non-significant p-value of 0.394, indicating no evidence of publication bias among studies included in this review (Table 2).

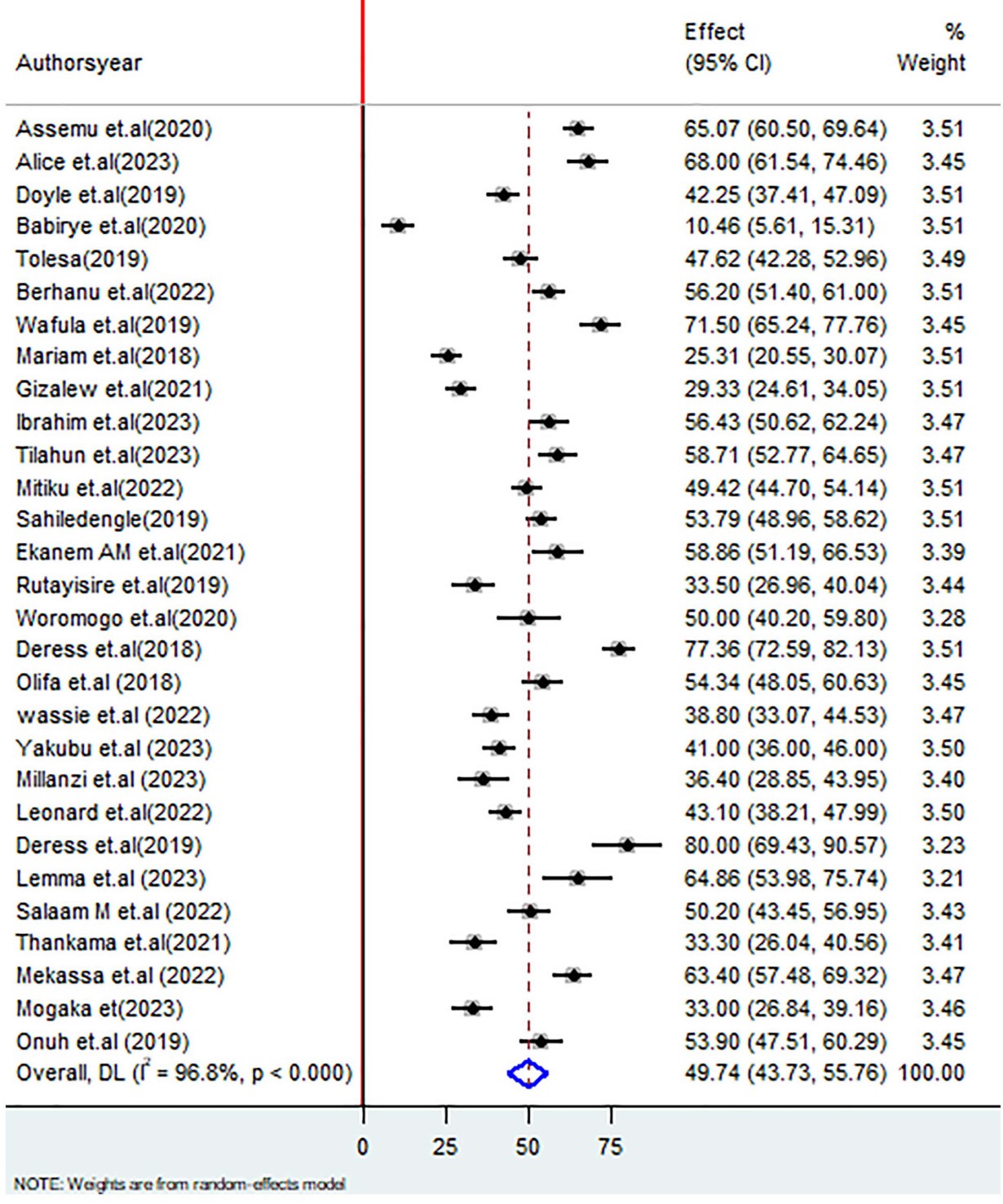

**Fig 2. Forest plot on pooled estimate of healthcare waste management practices among healthcare workers in SSA.**

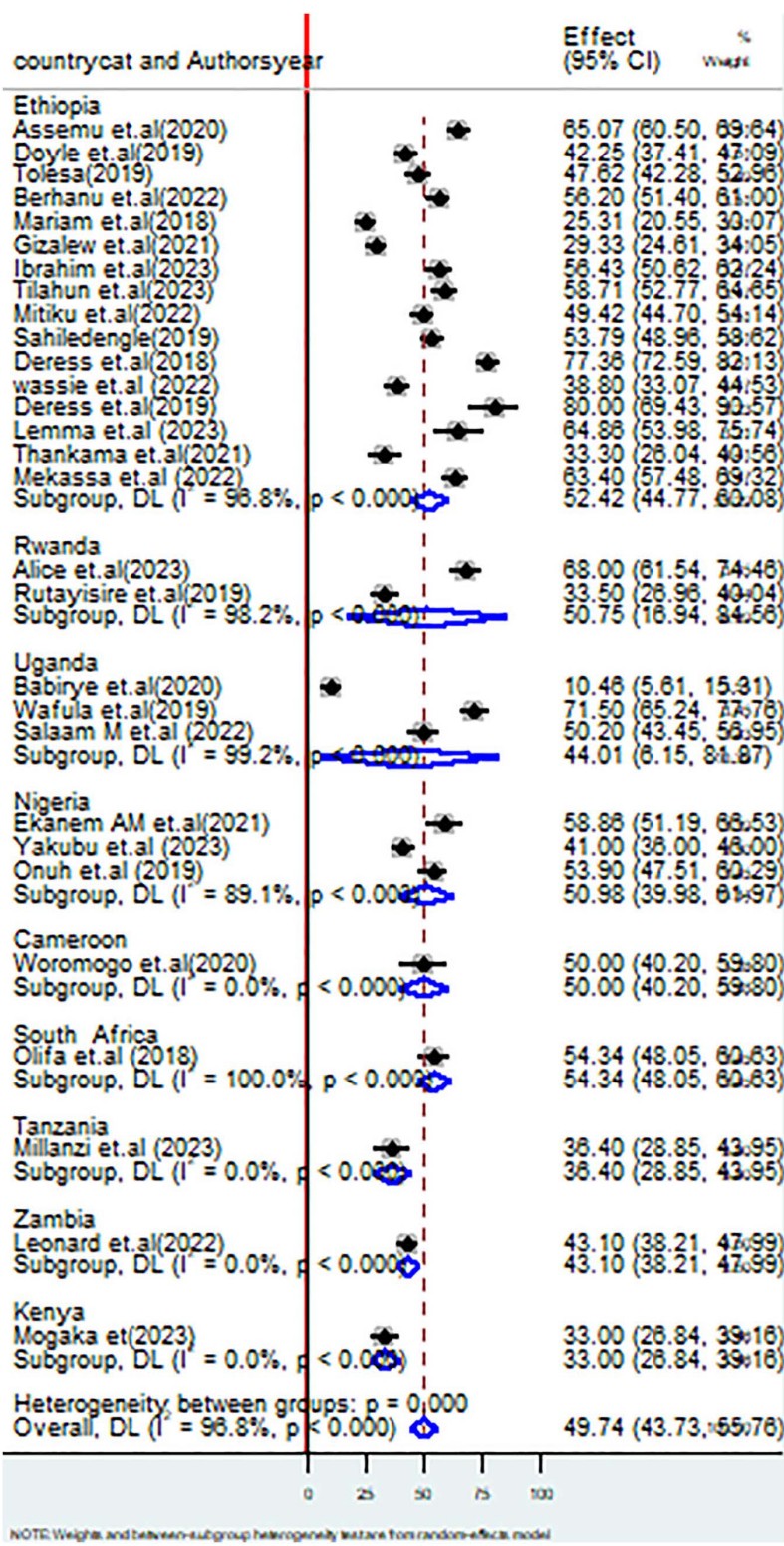

**Fig 3. Forest plot on subgroup analysis based on country on healthcare waste management practices among healthcare workers in SSA.**

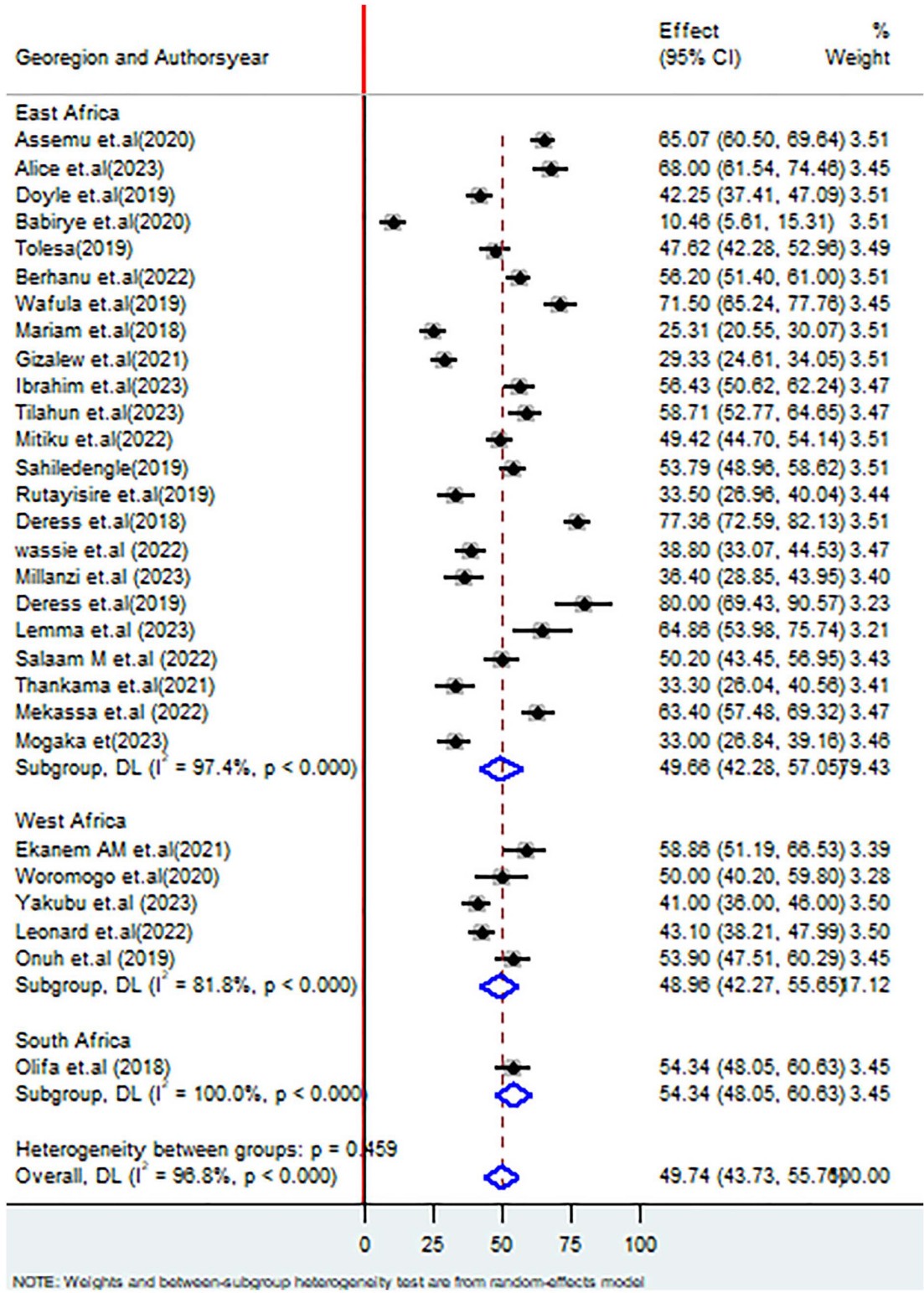

**Fig 4. Forest plot on subgroup analysis based on geographic region on healthcare waste management practices among healthcare workers in SSA.**

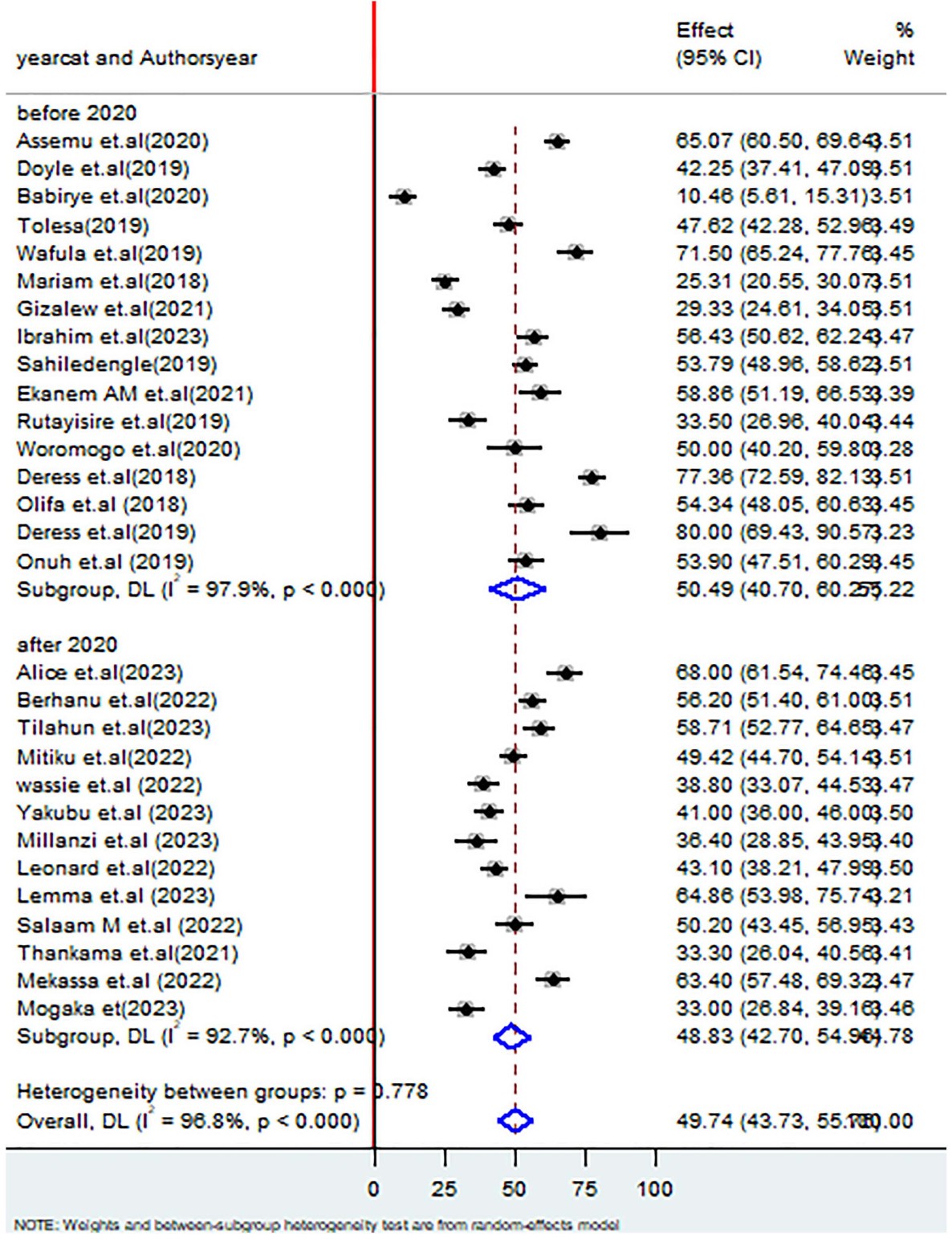

**Fig 5. Forest plot on subgroup analysis based on study year on healthcare waste management practices among healthcare workers in SSA.**

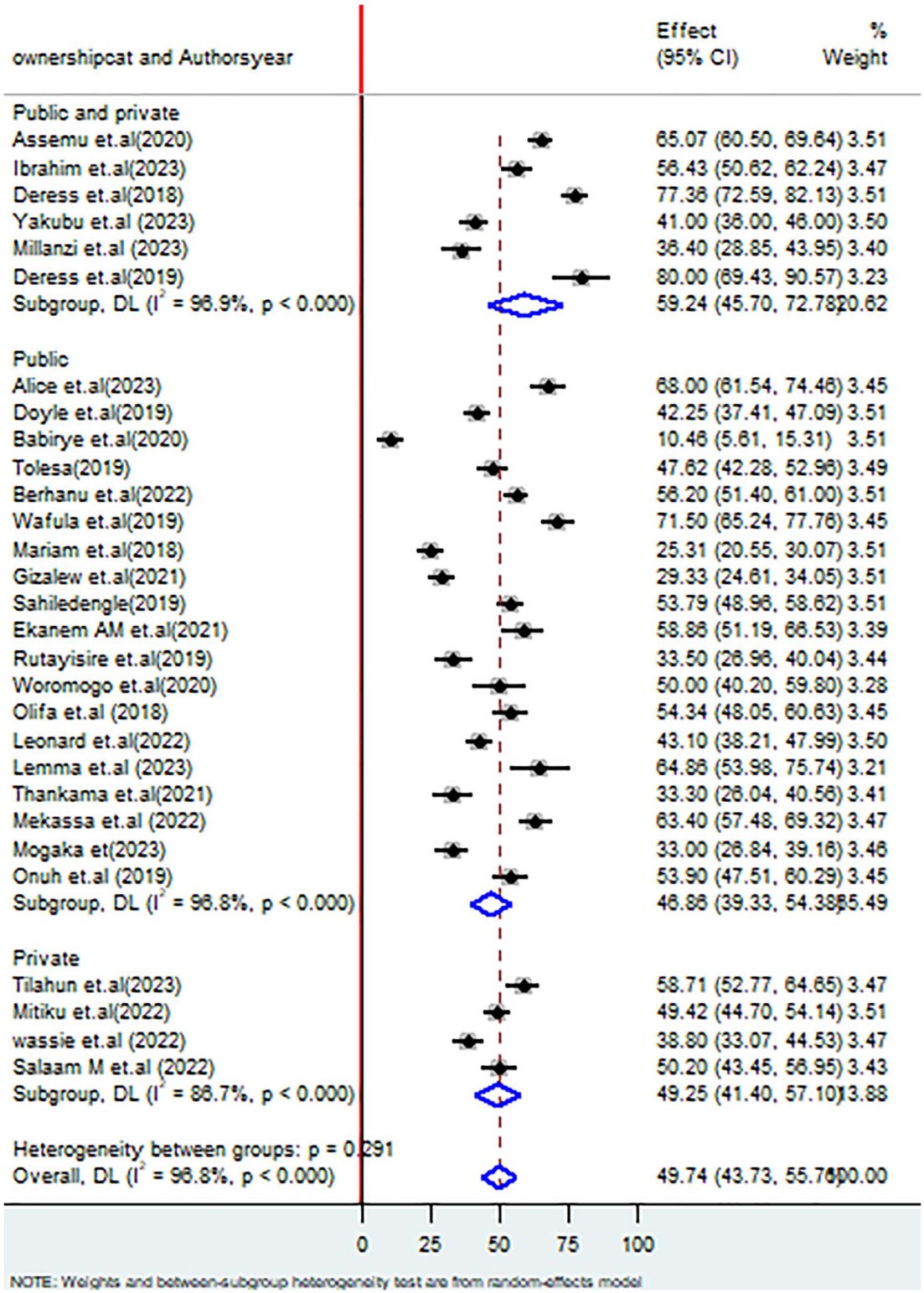

**Fig 6. Forest plot on subgroup analysis based on ownership of the healthcare facilities on healthcare waste management practices among healthcare workers in SSA.**

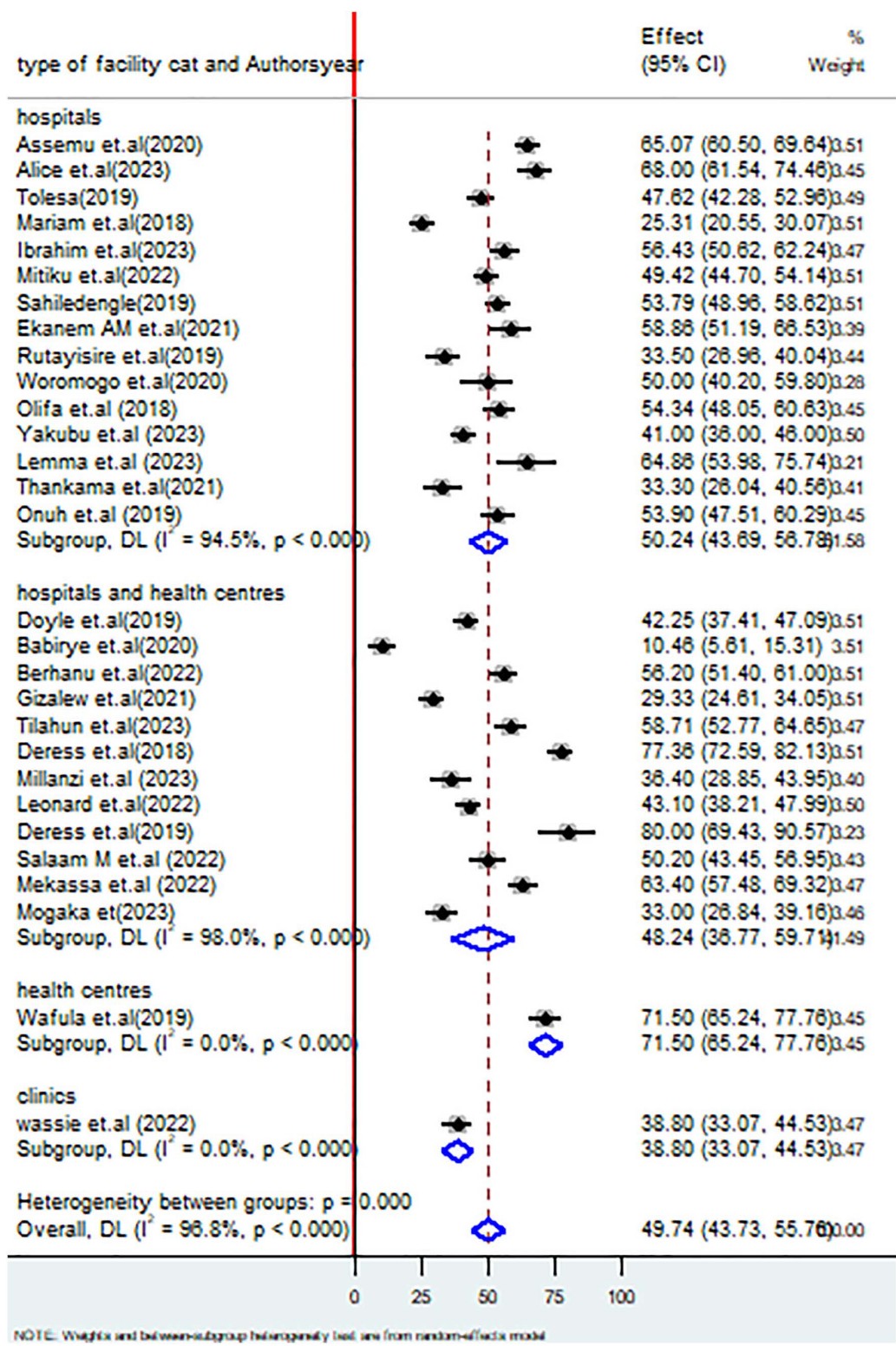

**Fig 7. Forest plot on subgroup analysis based on types of healthcare facilities on healthcare waste management practices among healthcare workers in SSA.**

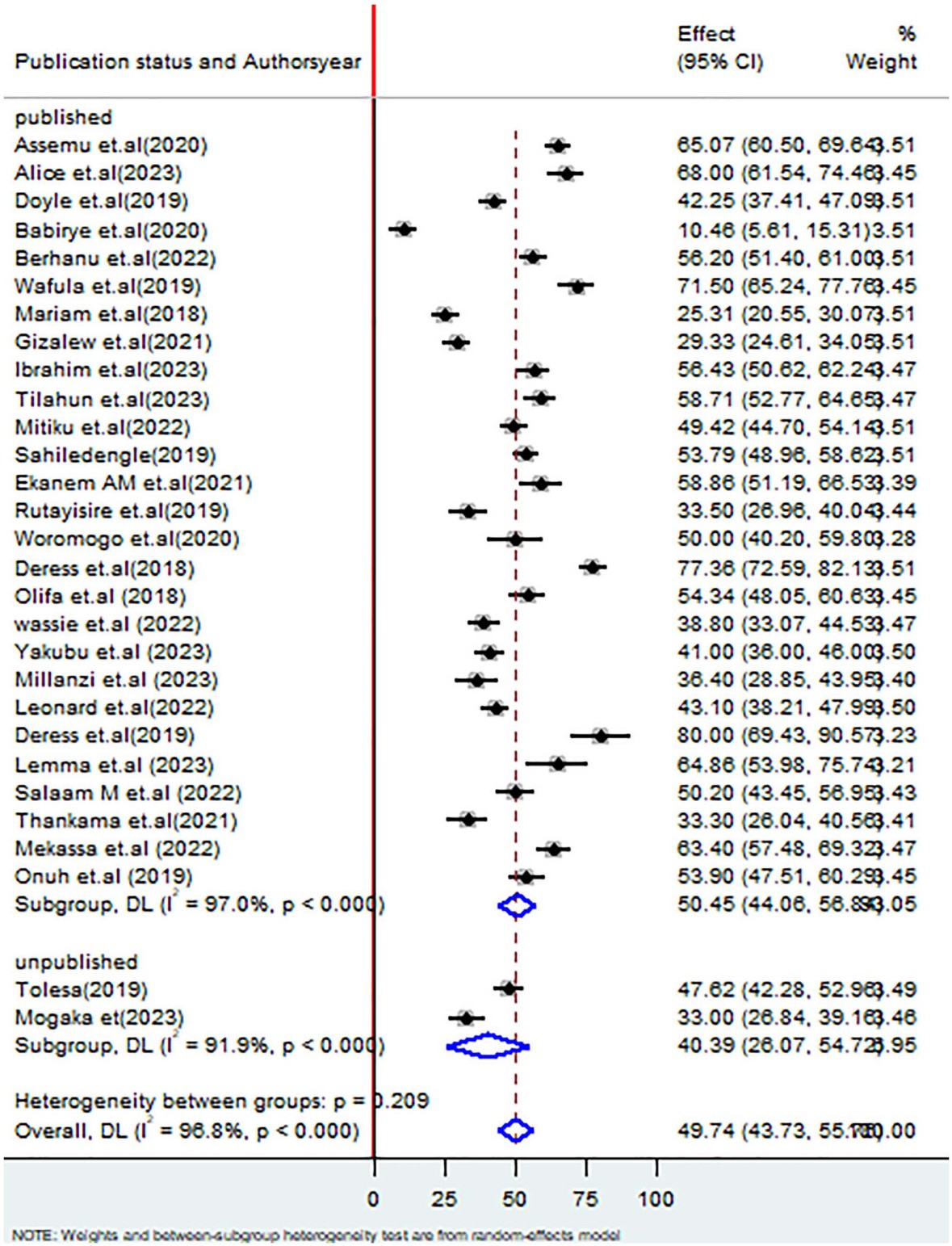

**Fig 8. Forest plot on subgroup analysis based on publication status on healthcare waste management practices among healthcare workers in SSA.**

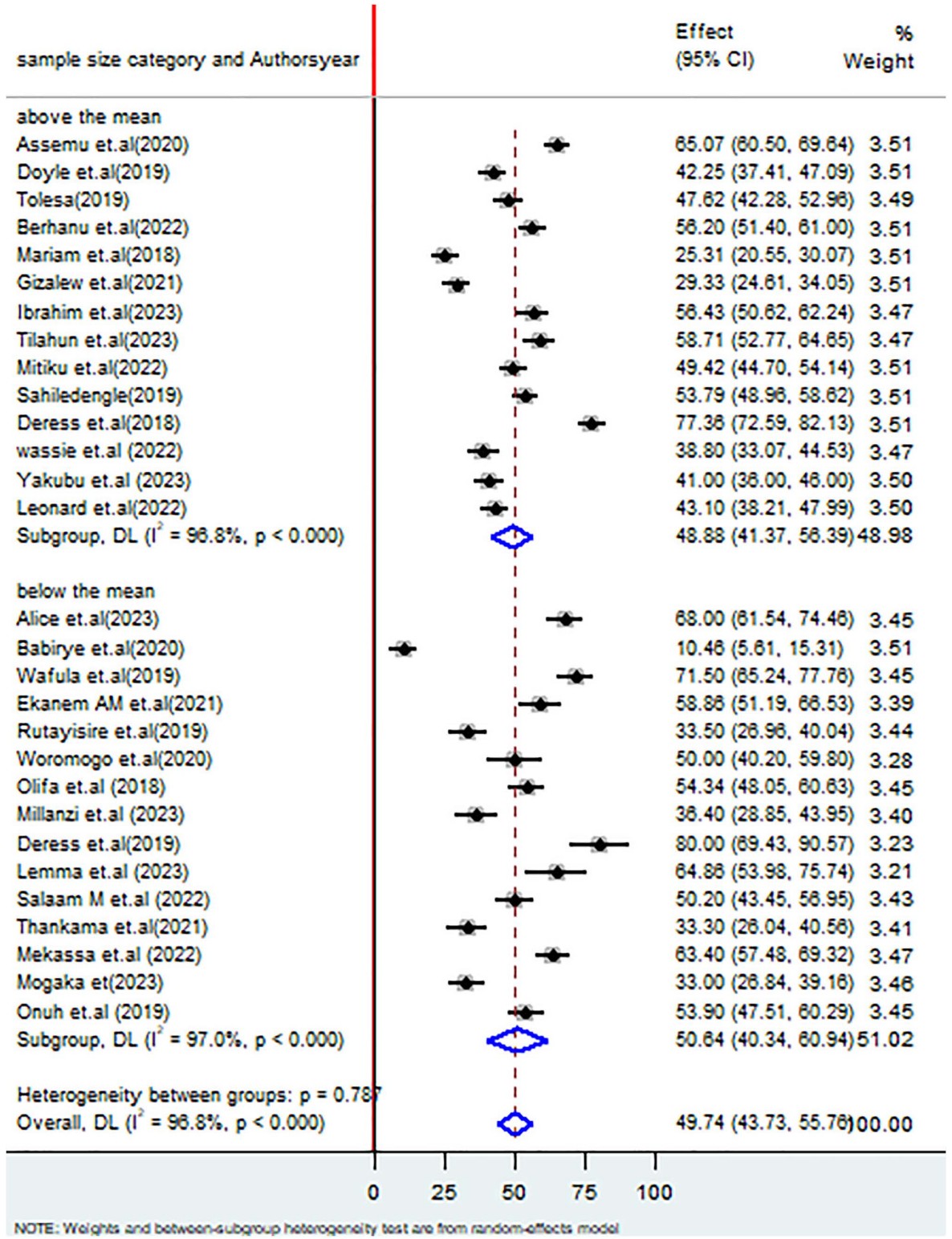

| sample size category and Authorsyear | Effect (95% CI) | % Weight |
|---|---|---|
| **above the mean** | | |
| Assemu et.al(2020) | 65.07 (60.50, 69.64) | 3.51 |
| Doyle et.al(2019) | 42.25 (37.41, 47.09) | 3.51 |
| Tolesa(2019) | 47.62 (42.28, 52.96) | 3.49 |
| Berhanu et.al(2022) | 56.20 (51.40, 61.00) | 3.51 |
| Mariam et.al(2018) | 25.31 (20.55, 30.07) | 3.51 |
| Gizalew et.al(2021) | 29.33 (24.61, 34.05) | 3.51 |
| Ibrahim et.al(2023) | 56.43 (50.62, 62.24) | 3.47 |
| Tilahun et.al(2023) | 58.71 (52.77, 64.65) | 3.47 |
| Mitiku et.al(2022) | 49.42 (44.70, 54.14) | 3.51 |
| Sahiledengle(2019) | 53.79 (48.96, 58.62) | 3.51 |
| Deress et.al(2018) | 77.36 (72.59, 82.13) | 3.51 |
| wassie et.al (2022) | 38.80 (33.07, 44.53) | 3.47 |
| Yakubu et.al (2023) | 41.00 (36.00, 46.00) | 3.50 |
| Leonard et.al(2022) | 43.10 (38.21, 47.99) | 3.50 |
| Subgroup, DL ($I^2$ = 96.8%, p < 0.000) | 48.88 (41.37, 56.39) | 48.98 |
| | | |
| **below the mean** | | |
| Alice et.al(2023) | 68.00 (61.54, 74.46) | 3.45 |
| Babirye et.al(2020) | 10.46 (5.61, 15.31) | 3.51 |
| Wafula et.al(2019) | 71.50 (65.24, 77.76) | 3.45 |
| Ekanem AM et.al(2021) | 58.86 (51.19, 66.53) | 3.39 |
| Rutayisire et.al(2019) | 33.50 (26.96, 40.04) | 3.44 |
| Woromogo et.al(2020) | 50.00 (40.20, 59.80) | 3.28 |
| Olifa et.al (2018) | 54.34 (48.05, 60.63) | 3.45 |
| Millanzi et.al (2023) | 36.40 (28.85, 43.95) | 3.40 |
| Deress et.al(2019) | 80.00 (69.43, 90.57) | 3.23 |
| Lemma et.al (2023) | 64.86 (53.98, 75.74) | 3.21 |
| Salaam M et.al (2022) | 50.20 (43.45, 56.95) | 3.43 |
| Thankama et.al(2021) | 33.30 (26.04, 40.56) | 3.41 |
| Mekassa et.al (2022) | 63.40 (57.48, 69.32) | 3.47 |
| Mogaka et(2023) | 33.00 (26.84, 39.16) | 3.46 |
| Onuh et.al (2019) | 53.90 (47.51, 60.29) | 3.45 |
| Subgroup, DL ($I^2$ = 97.0%, p < 0.000) | 50.64 (40.34, 60.94) | 51.02 |
| | | |
| Heterogeneity between groups: p = 0.787 | | |
| Overall, DL ($I^2$ = 96.8%, p < 0.000) | 49.74 (43.73, 55.76) | 100.00 |

NOTE: Weights and between-subgroup heterogeneity test are from random-effects model

**Fig 9. Forest plot on subgroup analysis based on sample size on healthcare waste management practices among healthcare workers in SSA in 2024.**

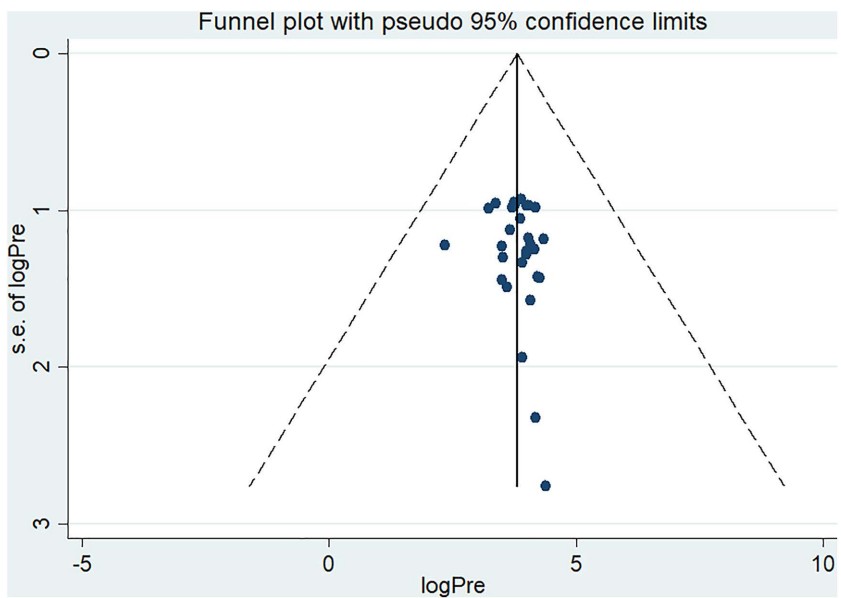

**Fig 10. Forest plot on funnel plot on publication bias facilities on healthcare waste management practices among healthcare workers in SSA.**

**Table 2. Eggers' regression test of studies included in on healthcare waste management practices among healthcare workers in SSA in 2024. Egger's test for small-study effects: Regress the standard normal deviate of the intervention. Effect estimate against its standard error. Number of studies = 29 Root MSE = 0.3354.**

| Std_Eff | Coef | Std. Err | T | p>t | [95%conf. Interval] | |
|---|---|---|---|---|---|---|
| Slope | 3.522179 | .3329556 | 10.58 | 0.000 | 2.839011 | 4.205348 |
| Bias | .2441147 | .2816182 | 0.87 | 0.394 | −.3337182 | .8219476 |

Test of Ho: no small-study effects P = 0.394.

**Sensitivity analysis.** The sensitivity analysis revealed that no single study unduly influenced the pooled estimate of good health healthcare waste management practices, confirming the robustness of the overall findings (Fig 11).

**Meta-regression analysis.** The heterogeneity assessment indicated substantial variation across the studies included in this review ($I^2 = 96.8\%$, $p < 0.001$), among the included studies. The meta-regression analysis was performed for Publication status, study year, country, geographic region, ownership of the healthcare facility, types of healthcare facilities, and sample size. The publication status has a regression coefficient of −0.276 with a p-value of 0.913, indicating a slight negative relationship with the effect size of healthcare waste management practices. However, this association is not statistically significant because the p-value exceeds 0.05. The wide confidence interval ranging from −5.45 to 4.90 reflects considerable uncertainty. Similarly, the study year presents a regression coefficient of 0.331 and a p-value of 0.801, suggesting a minor positive link with healthcare waste management practices, but this too is not statistically significant given the broad confidence interval from −2.36 to 3.02. Furthermore, the country variable has a regression coefficient of −0.119 with a p-value of 0.772, implying a slight negative effect on healthcare waste management practices but without significantly influencing heterogeneity. The geographic region shows a regression coefficient of 0.412 and a p-value of 0.815, indicating a positive association that is not statistically significant. Ownership of healthcare facilities has a regression coefficient of −0.249 and a p-value of 0.822, suggesting a negative association, but there is no strong evidence to support this.

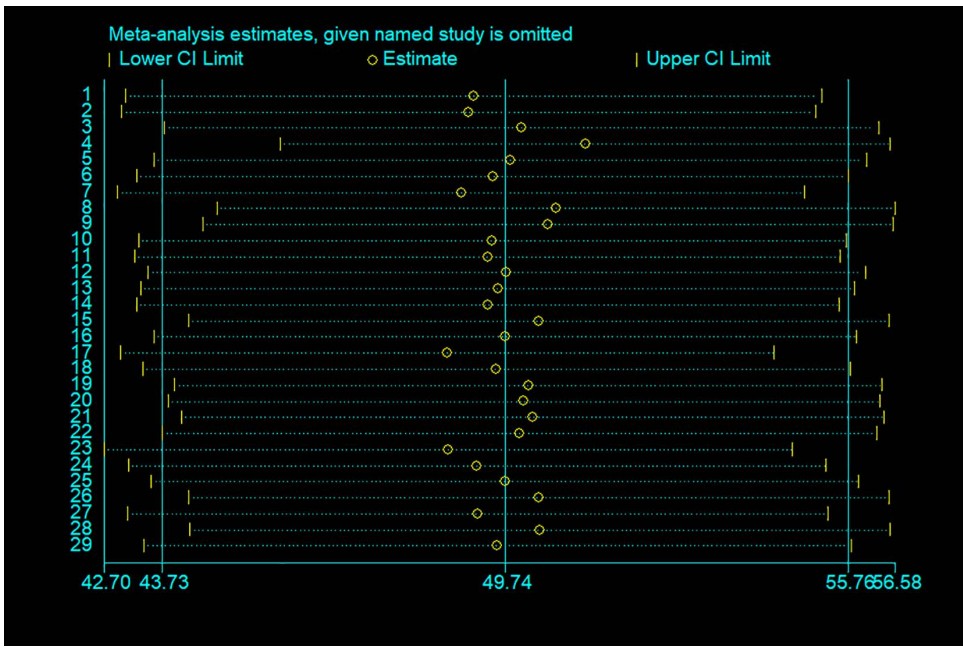

**Fig 11. Sensitivity analysis of the pooled proportion of healthcare waste management practices among healthcare workers in SSA.**

The type of healthcare facilities was analyzed as a potential heterogeneity source and showed a regression coefficient of 0.044 with a p-value of 0.961, reflecting a weak positive association without statistical significance. Finally, the sample size has a regression coefficient of −0.058 and a p-value of 0.963, indicating a small negative association, but not statistically significant. The constant term in the regression analysis is 4.237, representing the estimated pooled effect size when all predictor variables are zero. Although this value is positive, it is not statistically significant because its p-value is greater than 0.05. Generally, none of the moderators included in this meta-regression significantly explain the heterogeneity observed in healthcare waste management practices across the studies, as all p-values exceed the 0.05 threshold level. The confidence intervals for all variables are broad and cross zero, indicating uncertainty and insufficient evidence for these moderators as meaningful predictors. This suggests that the variation in the pooled effect size cannot be adequately explained by the variables tested, and other unmeasured subtle factors may contribute to the sources of heterogeneity (Table 3).

**Factors associated with healthcare waste management practices.** A total of ten studies [1,8,9,31,36,38,43–45,48] were included to assess factors associated with HCWM practices among healthcare workers. Three studies [1,44,45] revealed that male healthcare workers were 1.76 times more likely to implement good HCWM practices than female workers (OR: 1.76, 95% CI: 1.01–2.51). Additionally, three studies [1,31,38] revealed that healthcare workers who used manuals or guidelines in their working departments were 3.17 times more likely to implement good HCWM practices than those who did not use the guidelines (OR: 3.17, 95% CI: 1.58–4.75). Furthermore, three studies [31,38,43] revealed that healthcare workers with good knowledge on HCWM practices were 2.56 times more likely to implement good HCWM practices than those with poor knowledge of HCWM practices (OR: 2.56, 95% CI: 1.42–3.69). Six studies [8,9,36,38,43,48] showed that healthcare workers who had received training on HCWM were 1.29 times more likely to implement good HCWM practices compared to those who did not receive training (OR: 1.29, 95% CI: 1.09–1.49). Finally, two studies [8,38] revealed that healthcare workers who worked 8 hours or less per day were 4.98 times more likely to have good HCWM practices compared to those who worked for more than 8 hours per day (Fig 12).

**Table 3. Meta-regression analysis to assess the pooled estimate of healthcare waste management practices and associated factors among healthcare workers in Sub-Saharan Africa.**

| Variables | Coefficient | Std. error | 95% CI | p-value |
|---|---|---|---|---|
| Publication status | −0.2758118 | 2.487726 | −5.449321,4.897698 | 0.913 |
| Study year | .3307225 | 1.293202 | −2.358638, 3.020083 | 0.801 |
| Country | −.1193956 | .4071663 | −.9661444,.7273531 | 0.772 |
| Geographic region | .4118157 | 1.739553 | −3.205782, 4.029414 | 0.815 |
| Ownership of the HCFs | −.2486345 | 1.092535 | −2.520685, 2.023416 | 0.822 |
| Types of HCFs | .0435428 | .8858338 | −1.798649, 1.885737 | 0.961 |
| Sample size | −.0578419 | 1.243137 | −2.643087, 2.527403 | 0.963 |
| Constant | 4.236827 | 2.953837 | −1.906012, 10,37967 | 0.116 |

## Discussion

The growing global concern has brought to the forefront improper healthcare waste management practices in developing countries across the globe [57]. Poor healthcare waste management practice is still a major threat to environmental and public health, particularly in developing countries. Hence, proper healthcare waste management should be implemented strictly to alleviate these issues. The mismanagement of healthcare waste can negatively impact healthcare workers, communities, and clients of the healthcare facilities, and the communities as a whole [4]. Hence, this study aims to assess healthcare waste management practices and associated factors among healthcare workers in Sub-Saharan Africa.

This systematic review and meta-analysis focuses on Sub-Saharan Africa, a large and diverse geographic area where substantial variation among the included studies is anticipated, in terms of healthcare waste management practices. Literature revealed that a fixed effect model is more appropriate when there is lower heterogeneity, usually less than 25%, because it assumes that the true effect size is the same across all studies. However, in our case, the analysis of heterogeneity observed among the studies was very high, with an I² statistic reaching 96.8%. Due to this pronounced level of heterogeneity, it is more suitable to apply a random effects model, which accounts for and evaluates the variability in effect sizes across the different studies [58]. The finding of this review revealed that the pooled estimate of good healthcare waste management practices among healthcare workers in SSA was 49.74% (95% CI: 43.73, 55.76%), I² 96.8%, P<0.000 which was supported with studies done in Pakistan 47% [59], a systematic review and meta-analysis in Ethiopia (52.86%) [60], Saudi Arabia (49.5%) [7], and Indonesia (57.94%) [61]. On the other hand, the finding of this systematic review and meta-analysis was lower than the studies done in Pakistan (65%) [62], India (81.75%) [63], 69.33% [64], (89.6%) [65], United Kingdom (90%) [66], and Thailand (92.2%). On the contrary, the finding of this review was higher than the study in Tunisia (25%) [67]. These disparities may stem from gaps in knowledge, poor waste segregation, absence of policies, inadequate planning and training, limited awareness of health risks, weak infrastructure, and insufficient treatment technologies [68]. Additionally, this disparity may be explained due to a variation in sample size, study settings, and the willingness and commitment of healthcare workers to adopt such practices. The COVID-19 pandemic exacerbated the existing healthcare waste management during the COVID-19 pandemic have faced several challenges due to increased production of infectious waste, interruption of recycling strategy, and inadequate resources to handle increased waste production [7]. Subgroup analysis by study year, comparing periods before and after the COVID-19 pandemic, indicated that during the pandemic, a higher proportion of healthcare workers demonstrated good healthcare waste management practices was 50.49% (95% CI: 40.70,60.25%), whereas after the occurrences of the pandemic was 48.83% (95% CI: 42.70,54.96%). Despite the increased focus on infection control and proper waste handling protocols following the COVID-19 pandemic, challenges in healthcare waste management practices persist as usual, which may be due to the diverse characteristics and bulk quantity of wastes generated in the healthcare facilities [69]. Furthermore, Post-pandemic, budget limitations may hinder the adequate provision of personal protective equipment [70].

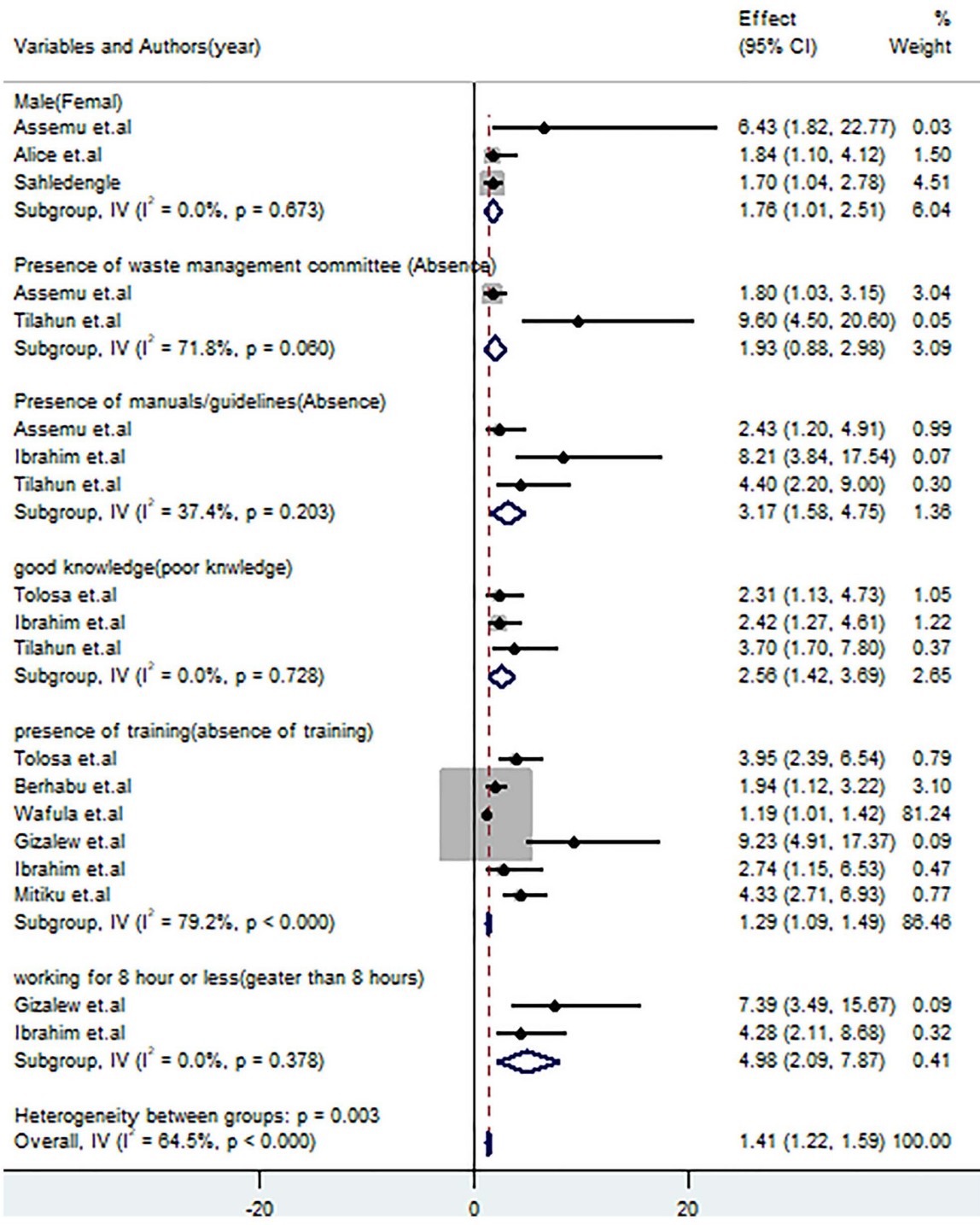

**Fig 12. Forest plot on factors associated with healthcare waste management practices among healthcare workers in SSA.**

Regarding the ownership of healthcare facilities, this review revealed that healthcare workers, who worked in public healthcare facilities, had 46.86% (39.33, 54.38% whereas those who worked in private healthcare facilities had 49.25% (95% CI: 41.40, 57.10%). Poor healthcare waste management in public facilities may result from high patient volume, limited resources, weak monitoring systems, and inadequate segregation and treatment infrastructure. The lack of infrastructure and funding for waste management in public healthcare settings may be a key driver of the lower proportion of good practices [14,71,72]. Furthermore, funding restrictions and a lack of government commitment may restrict the opportunities for implementing good healthcare waste management practices [14]. This review also came up with factors significantly associated with healthcare waste management practices among healthcare workers in SSA. These factors include the workers' sex, the availability of healthcare waste management manuals or guidelines in their working departments, knowledge on proper waste management practices, the status of training on healthcare waste management, and the number of working hours per day. The finding of this review revealed that male healthcare workers were 1.76 (1.01, 2.51) times more likely to implement good healthcare waste management practices than females, which was supported by a study done in Saudi Arabia [7]. This situation may be attributed to domestic responsibilities among female workers, which could affect their effectiveness in the workplace.

Comprehensive training for healthcare workers is pivotal for enhancing the knowledge gap, securing worker buy-in, and ultimately driving improved waste management practices in healthcare settings, which is a critical factor for the successful healthcare waste management [73]. However, most of the healthcare workers did not receive sufficient infection prevention and control training, which may affect their knowledge status towards implementing proper healthcare waste management practices [7]. This review concluded that healthcare workers who received training on HCWM practices were 1.29 times more likely to implement good waste management practices compared with workers who did not receive the training, which was supported by a studies done in Ethiopia [60], India [65,74], and the United Kingdom [66]. This may be explained by the fact that trained healthcare workers enable them to utilize standard guidelines and procedures for implementing standard healthcare waste management practices. This finding highlights the essential of pre-service and in-service training on infection prevention and control practices for healthcare workers. The knowledge of healthcare workers towards healthcare waste management practices plays a vital role in promoting good hygiene, raising public awareness, and upholding responsible waste management practices across the entire healthcare system [75]. Their understanding is also key to mitigate occupational and community health risks associated with improper healthcare waste handling [73]. The finding of this review disclosed that healthcare workers who have good knowledge towards healthcare waste management practices had 2.56 times more likely to implement good waste management practices than compared with those having poor knowledge which was matched with a systematic review and meta-analysis done in Ethiopia [60], Bangladesh [76], Pakistan [62], India [65], Indonesia [61], and China [29,75]. Therefore, addressing awareness, infrastructure, resources, and organizational problems may enhance good HCWM practices [73,77].

This review also came up with healthcare waste management practices among healthcare workers were influenced by the availability of working guidelines/manuals in their healthcare settings. Healthcare workers who utilized these guidelines were 3.17 (1.58, 4.75) times more likely to implement good healthcare waste management practices than their counterparts, which was supported by a study done in Pakistan [78]. Additionally, healthcare workers who worked for shifts of 8 hours or less per day were 4.98 (2.09, 7.87) times more likely to implement good healthcare waste management practices than those who worked more than eight hours per day, which was supported by a study done in Indonesia [61]. Healthcare workers who are overwhelmed with their regular duties may struggle to effectively carry out their daily tasks, including proper healthcare waste management practices. Sub-group analysis and meta-regression were conducted to determine the sources of heterogeneity. However, neither of them identified the sources of heterogeneity. The variations in cultural, social, and environmental contexts may limit the generalizability of findings, posing a risk of context-specific conclusions and misleading recommendations [79].

## Strengths and limitations of the study

This systematic review and meta-analysis have certain limitations. Firstly, although the review was conducted through a thorough review of literature across multiple databases using predefined sets of criteria to reduce selection bias, some limitations remain. The high level of heterogeneity among studies included in this review was ($I^2 = 96.8\%$), which may complicate the ability to draw definitive conclusions and overgeneralization. Additionally, limiting the eligibility criteria to studies published in English could have led to the exclusion of pertinent findings done in other languages. Another limitation of this review is the exclusion of qualitative studies, which might have offered valuable insights into healthcare waste management practices, policies, and strategies. Furthermore, this review includes some preprint articles that have not undergone peer review, which could introduce bias due to the methodologies employed, and thus, the findings may need to be revised as more evidence emerges in the future.

## Conclusions and recommendations

The findings of this systematic review and meta-analysis revealed that only half of the healthcare workers implement good healthcare waste management practices in SSA. The finding also concluded that despite there being heterogeneity among studies included in this review, with a value of $I^2 = 96.8\%$, the subgroup analysis revealed that there was no statistical difference among studies. Meta-regression analysis was conducted to investigate the sources of heterogeneity, revealing that some factors contributed to the variability, which suggests that there may be other unidentified sources of heterogeneity. In terms of influencing factors, the sex of healthcare workers, availability of healthcare waste management manuals or guidelines in their departments, healthcare workers' knowledge about proper waste management practices, the presence of training on waste management, and the number of working hours per day were all significantly associated with healthcare waste management practices among healthcare workers. Hence, healthcare workers should receive regular in-service training, with particular emphasis on female staff. Additionally, concerned governmental and non-governmental health organizations should closely oversee and ensure daily adherence to healthcare waste management practices, as well as ensure the availability of appropriate healthcare waste management practices manuals and guidelines.

## Supporting information

**S1 File. Prisma checklist.**
(DOCX)

**S2 File. JBI quality assessment of studies included in this review.**
(DOCX)

**S3 File. Data.**
(XLSX)

## Acknowledgments

We sincerely acknowledge Debre Markos University for facilitating internet access, which enabled us to retrieve essential literature for this review. We also deeply appreciate the valuable input and support from our colleagues and friends from Prospero registration to manuscript preparation.

## Author contributions

**Conceptualization:** Gete Berihun, Zebader Walle, Belay Desye, Chala Daba, Abebe Kassa Geto, Lake Kumlachew, Leykun Berhanu.

**Data curation:** Gete Berihun, Zebader Walle, Belay Desye, Chala Daba, Abebe Kassa Geto, Lake Kumlachew, Leykun Berhanu.

**Formal analysis:** Gete Berihun, Zebader Walle, Belay Desye, Chala Daba, Lake Kumlachew.

**Funding acquisition:** Gete Berihun.

**Investigation:** Gete Berihun, Zebader Walle, Belay Desye, Chala Daba, Abebe Kassa Geto, Lake Kumlachew, Leykun Berhanu.

**Methodology:** Gete Berihun, Zebader Walle, Belay Desye, Chala Daba, Abebe Kassa Geto, Lake Kumlachew, Leykun Berhanu.

**Project administration:** Gete Berihun, Leykun Berhanu.

**Resources:** Gete Berihun, Zebader Walle, Belay Desye, Lake Kumlachew.

**Software:** Gete Berihun, Zebader Walle, Belay Desye, Chala Daba, Abebe Kassa Geto, Lake Kumlachew, Leykun Berhanu.

**Supervision:** Gete Berihun, Zebader Walle, Belay Desye, Chala Daba, Abebe Kassa Geto, Lake Kumlachew.

**Validation:** Gete Berihun, Belay Desye, Abebe Kassa Geto, Lake Kumlachew, Leykun Berhanu.

**Visualization:** Gete Berihun, Zebader Walle, Chala Daba, Abebe Kassa Geto, Leykun Berhanu.

**Writing – original draft:** Gete Berihun, Zebader Walle, Belay Desye, Chala Daba, Abebe Kassa Geto, Lake Kumlachew, Leykun Berhanu.

**Writing – review & editing:** Gete Berihun, Zebader Walle, Belay Desye, Chala Daba, Abebe Kassa Geto, Lake Kumlachew, Leykun Berhanu.

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
