## [Decision Letter · Decision Letter 0]

17 Dec 2024

PONE-D-24-32196Healthcare waste management practice and associated factors among healthcare workers in Sub-Saharan Africa:  a systematic review and meta-analysis.PLOS ONE

Dear Dr. Berihun,

Thank you for submitting your manuscript to PLOS ONE. After careful consideration, we feel that it has merit but does not fully meet PLOS ONE’s publication criteria as it currently stands. Therefore, we invite you to submit a revised version of the manuscript that addresses the points raised during the review process.

Dear authors try to address the comments given by reviewer 1 and 2 as their decision is major reviesion.

We look forward to receiving your revised manuscript.

Kind regards,

Tsegaye Alemayehu, Msc

Academic Editor

PLOS ONE

Journal Requirements:

2. We note that there is identifying data in the Supporting Information file < S2.docx>. Due to the inclusion of these potentially identifying data, we have removed this file from your file inventory. Prior to sharing human research participant data, authors should consult with an ethics committee to ensure data are shared in accordance with participant consent and all applicable local laws.

-Location data

Additional guidance on preparing raw data for publication can be found in our Data Policy (https://journals.plos.org/plosone/s/data-availability#loc-human-research-participant-data-and-other-sensitive-data ) and in the following article: http://www.bmj.com/content/340/bmj.c181.long .

Please remove or anonymize all personal information (Name), ensure that the data shared are in accordance with participant consent, and re-upload a fully anonymized data set. Please note that spreadsheet columns with personal information must be removed and not hidden as all hidden columns will appear in the published file.

3. Please include captions for your Supporting Information files at the end of your manuscript, and update any in-text citations to match accordingly. Please see our Supporting Information guidelines for more information: http://journals.plos.org/plosone/s/supporting-information .

Reviewers' comments:

Reviewer's Responses to Questions

**Comments to the Author**

1. Is the manuscript technically sound, and do the data support the conclusions?

Reviewer #1: Yes

Reviewer #2: Partly

Reviewer #3: Yes

2. Has the statistical analysis been performed appropriately and rigorously? 

Reviewer #1: Yes

Reviewer #2: No

Reviewer #3: Yes

3. Have the authors made all data underlying the findings in their manuscript fully available?

Reviewer #1: Yes

Reviewer #2: Yes

Reviewer #3: Yes

4. Is the manuscript presented in an intelligible fashion and written in standard English?

Reviewer #1: Yes

Reviewer #2: No

Reviewer #3: Yes

5. Review Comments to the Author

Reviewer #1: Date: 15 September 2024

Manuscript Title: Healthcare waste management practice and associated factors among healthcare workers in Sub-Saharan Africa: a systematic review and meta-analysis

Journal: PLOS ONE

Manuscript ID: PONE-D-24-32196

Manuscript type: Review article

GENERAL COMMENTS:

Authors reviewed a topic which is highly relevant for healthcare workers, practitioners, regulators, and policy makers regarding healthcare waste management. The methodology and results are very good. However, the articles included in the review are mainly from East Africa/Ethiopia. Hence, further literature search and analysis is required if the term SSA is included in the title of the manuscript. Therefore, the manuscript can be accepted for publication after major revision.

SPECIFIC COMMENTS:

Introduction:

1. Authors cited local studies for global figures which is not appropriate. For example, line 61-63: “Annually, 5.2 million people worldwide, including 4 million children, death are reported as a result of improper management [7]. Additionally, more than 2 million healthcare workers are infected by over 30 dangerous blood - borne pathogens [4]. Reference number 4 and 7 are a study from Ethiopia but cited for the global public health burden associated with poor healthcare waste management..

2. Check the validity of the statement from line 64-66: “Exposure to hazardous waste containing mercury and expired drugs can lead to serious health issues like cancer, mutations, teratogenicity, and infections of the eyes, respiratory system, and skin diseases [9]” How infection of the eye and the like from mercury and expired drugs exposure.

3. Line 69-74 discusses improper disposal of healthcare waste such as indiscriminate dumping, burying and burning as well as the associated health and environmental risks. Thes practices are not related to healthcare workers. These issues are applicable for management bodies and medical waste handlers/janitors. It’s better to remove this part.

4. The introduction lacks healthcare workers waste management practice information. Authors reviewed mainly risks associated with poor waste management practices.

Materials and Methods:

5. On page 4, Line 106: Delete the subtitle “Study Setting: The study setting is SSA.” As it is already indicated in the manuscript title and in line 92-93 on the same page.

6. Among 18 articles included in the meta-analysis, more than 60% were from Ethiopia. When articles from Ethiopia, Ruanda and Uganda combined, they account for 83% from the total articles in the SSA countries. Is it appropriate to mention SSA in the title? If it appears in the title, more articles are required from other SSA countries. Authors compared their review result with the study conducted in Kenya [“………… the finding of this review was lower than the studies done in Kenya (72.5%) [43]”] but did not include it in the review. Why?

Country Number of included articles %

Ethiopia 11 61.1

Ruanda 2 11.1

Uganda 2 11.1

East Africa 15 83.3

Nigeria 1 5.6

Cameron 1 5.6

South Africa 1 5.6

Total 18 100

7. Authors reach a conclusion based on the ‘Good Practice’ report of each included study. However, the operational definition of ‘Good Practice’ might not be similar for all studies. This will have implications for the validity and reliability of the method to measure the outcome variable. This need clarification.

Discussion:

8. Page 14: “This review also revealed that the sex of the healthcare workers, the presence of healthcare waste management manuals/ guidelines in their working departments, the knowledge status of the healthcare workers, the presence of training on healthcare waste management practices, and working hours per a day were factors significantly associated with healthcare waste management practices among healthcare workers.” Is it a result or a discussion?

9. In the last paragraph: though the findings are compared with similar study results, justification for favoring or discouraging good practices were not mentioned.

10. No comparison is made with studies from North African countries, Latin America countries and the like.

Conclusions and Recommendations:

11. “This review revealed that the pooled estimated good practices of HCWM among healthcare workers in SSA were relatively low.” What does ‘relatively low mean’?

12. “The sex of the healthcare worker, the presence of waste management manuals/guidelines, knowledge of healthcare workers on healthcare waste management, training on waste management, and the working hours per day were factors significantly associated with HCWM practices among healthcare workers.“ Is it being male or female that tends to good practice? Is working less than or more than 8 hours that favors good practice? Show the direction of each factor towards good practice.

Reviewer #2: Thank you for submitting your manuscript titled "Healthcare waste management practice and associated factors among healthcare workers in Sub-Saharan Africa: a systematic review and meta-analysis." Below, I provide detailed feedback based on my evaluation of the manuscript.

The manuscript is based on a systematic review and meta-analysis, which is an appropriate method for synthesizing data on healthcare waste management (HCWM) practices in Sub-Saharan Africa. The methodology is described in detail, including adherence to PRISMA guidelines, the use of statistical software (STATA), and inclusion criteria based on robust study designs.

The conclusions are partly supported by the data but are limited by the following:

• The pooled estimate of HCWM practices (50.42%) is based on studies with very high heterogeneity (I² = 97.7%). While subgroup analyses attempt to explore sources of heterogeneity, significant variability remains, limiting the generalizability of the results.

• Pooling data across diverse countries with varying healthcare systems and policies may mask important contextual factors influencing HCWM practices.

• The exclusion of qualitative studies limits a deeper understanding of barriers and facilitators to HCWM practices, which is critical for policy and practice recommendations.

Address the high heterogeneity more thoroughly by exploring meta-regression or other advanced statistical methods. Consider discussing the potential limitations of pooling data across diverse contexts in more depth.

Appropriate statistical techniques were used, including random-effects modeling, Egger’s regression test for publication bias, and sensitivity analyses. Subgroup analyses by country, healthcare facility ownership, and pre- versus post-COVID-19 pandemic periods add depth to the findings.

• Despite subgroup analyses, high heterogeneity persists, which undermines the reliability of the pooled estimates.

• The justification for choosing the random-effects model is not explicitly stated, and alternative approaches (e.g., mixed-effects models) are not considered.

• The authors could have performed a meta-regression to explore potential moderators contributing to the observed heterogeneity.

Provide a clearer rationale for model selection and explore advanced methods to account for heterogeneity. This would enhance the robustness of the findings.

The authors have stated that all relevant data are fully available within the manuscript and supporting information. References to included studies and the search strategy are provided, ensuring transparency and compliance with PLOS ONE’s data policy.

The manuscript communicates the main findings and methodology effectively.

There are several grammatical errors, typographical mistakes, and instances of unclear phrasing that impact the readability and professionalism of the manuscript. Examples include:

• Overly complex sentences, such as “The pooled estimate of good HCWM practices among study participants was found to be 50.42%...” These could be simplified for clarity.

• Repetitive phrases, such as “good healthcare waste management practices,” could be varied to improve flow.

• Ambiguities in phrasing, such as “may cause the missing of some important studies,” should be revised to “may result in the omission of important studies.”

The manuscript requires editing for grammar, syntax, and readability to meet the standard of clear and unambiguous English required by PLOS ONE. A professional language editing service may be beneficial.

The manuscript outlines some strengths, such as the use of PRISMA guidelines, comprehensive literature search, and robust inclusion criteria. However:

• The exclusion of non-English studies may introduce bias and limit the generalizability of findings, particularly in multilingual regions.

• The reliance on quantitative studies excludes valuable qualitative insights that could enrich understanding.

Expand the discussion of the study’s limitations, particularly regarding heterogeneity, language restrictions, and the exclusion of qualitative studies. This would provide a more balanced view of the study’s findings.

• No concerns regarding research ethics or dual publication were identified.

• The forest plots and PRISMA diagram are informative but could benefit from improved labeling to enhance clarity for readers.

• While the authors provide recommendations for improving HCWM practices, these could be strengthened by incorporating actionable steps tailored to specific contexts (e.g., low-resource settings).

This manuscript addresses an important public health issue and has potential for significant impact. However, improvements in addressing heterogeneity, refining the statistical approach, and enhancing the clarity of the language are essential. I encourage the authors to revise the manuscript based on the feedback provided.

Thank you for the opportunity to review this manuscript.

Reviewer #3: The manuscript is technically sound, with a robust statistical framework appropriate for the systematic review and meta-analysis.

1. Enhance the introduction by clearly articulating the novel contributions of the study, such as new insights into the factors influencing healthcare waste management practices in this specific regional context.

2. I recommend using abbreviations sparingly, only for frequently repeated terms, or providing an abbreviation guide with the full meanings.

3. The reported high heterogeneity (I² = 97.7%) suggests substantial variability in the study outcomes, which could impact the generalisability of the findings. A more detailed exploration of the sources of this heterogeneity could strengthen the conclusions.

4. Some grammatical phrases need revision to enhance clarity and readability, e.g., "However, during by the two independent reviewers, the issues were resolved through an evidence-based discussion.".

6. PLOS authors have the option to publish the peer review history of their article (what does this mean? ). If published, this will include your full peer review and any attached files.

**Do you want your identity to be public for this peer review?** For information about this choice, including consent withdrawal, please see our Privacy Policy .

Reviewer #1: **Yes: ** Taffere Addis

Reviewer #2: **Yes: ** Volkan Medeni

Reviewer #3: **Yes: ** Sumayya Al-Mansur

---

## [Author Response · Author response to Decision Letter 1]

7 Jul 2025

Rebuttal letter

Response to the editor

#1. When submitting your revision, we need you to address these additional requirements.

Response: We have incorporated all the requirements as per the standards of PLOS ONE.

#2. Please include captions for your Supporting Information files at the end of your manuscript, and update any in-text citations to match accordingly. Please see our Supporting Information guidelines for more information: http://journals.plos.org/plosone/s/supporting-information.

Response: Thank you very much, we have modified the revised version of the manuscript based on its requirements.

Reviewer 1

#1. The authors cited local studies for global figures, which is not appropriate. For example, line 61-63: “Annually, 5.2 million people worldwide, including 4 million children, death are reported as a result of improper management [7]. Additionally, more than 2 million healthcare workers are infected with over 30 dangerous blood-borne pathogens [4]. References numbers 4 and 7 are a study from Ethiopia, but cited for the global public health burden associated with poor healthcare waste management.

Response: Sorry for the confusion we have created. Hence, we have modified the references accordingly in the revised version of the manuscript.

#2. Check the validity of the statement from line 64-66: “Exposure to hazardous waste containing mercury and expired drugs can lead to serious health issues like cancer, mutations, teratogenicity, and infections of the eyes, respiratory system, and skin diseases [9]” How infection of the eye and the like from mercury and expired drugs exposure.

Response: Sorry for the confusion we have created. We have rechecked the validity of the statement from the references I got it and it is a valid statement. The statement focus on the effects of exposing on hazardous waste containing mercury.

#3. Line 69-74 discusses improper disposal of healthcare waste such as indiscriminate dumping, burying and burning as well as the associated health and environmental risks. These practices are not related to healthcare workers. These issues are applicable for management bodies and medical waste handlers/janitors. It’s better to remove this part.

Response: Thank you very much for your critical insights to improve the quality of the manuscript. Hence we have amended the revised manuscript based on your suggestion.

#4. The introduction lacks healthcare workers waste management practice information. Authors reviewed mainly risks associated with poor waste management practices.

Response: thank you very much for your concern. Hence, we have used additional references which add pertinent information on healthcare waste management practices.

#5. On page 4, Line 106: Delete the subtitle “Study Setting: The study setting is SSA.” As it is already indicated in the manuscript title and lines 92-93 on the same page.

Response: Sorry for the confusion we have created. Hence, we have removed the subtitle study setting in the revised manuscript.

#6. Among 18 articles included in the meta-analysis, more than 60% were from Ethiopia. When articles from Ethiopia, Rwanda and Uganda are combined, they account for 83% of the total articles in the SSA countries. Is it appropriate to mention SSA in the title? If it appears in the title, more articles are required from other SSA countries. Authors compared their review result with the study conducted in Kenya [“………… the finding of this review was lower than the studies done in Kenya (72.5%) [43]”], but did not include it in the review. Why?

Response: Sorry for the confusion we have created. We have tried our best to find out more literature done in various SSA. Regarding the inclusion of SSA in the title, one of the advantages of using systematic review and meta-analysis is the use of pre-determined inclusion criteria to reduce the burden of bias. We have tried to find out more literature in different countries across the continent to reduce your concern as much as possible. Various research studies were conducted on healthcare waste management practices among healthcare workers, but they failed to qualify the inclusion criteria. The other critical concern is the use of literature done in Kenya to compare the findings of this review, but it should be part of the review, hence, Kenya is already part of SSA. We have amended it in the revised manuscript.

#7. The authors reach a conclusion based on the ‘Good Practice’ report of each included study. However, the operational definition of ‘Good Practice’ might not be similar for all studies. This will have implications for the validity and reliability of the method to measure the outcome variable. This needs clarification.

Response: Thank you very much for your critical insight. To overcome the problem you have mentioned we have used literatures which operationalizes as having of good healthcare waste management practices whey the perform more than 50% of the recommended healthcare waste management practices when implementing greater than or equal to the mean score of the practice measures.

#8. Page 14: “This review also revealed that the sex of the healthcare workers, the presence of healthcare waste management manuals/ guidelines in their working departments, the knowledge status of the healthcare workers, the presence of training on healthcare waste management practices, and working hours per a day were factors significantly associated with healthcare waste management practices among healthcare workers.” Is it a result or a discussion?

Response: Thank you very much for your concern. The above statement is more of a result section. However, in discussions the there is a trend to present the overall finding in a precise manner as the introduction part of the discussion. If you believe that this is not important, we can omit it from the place

#9. In the last paragraph, though the findings are compared with similar study results, justification for favoring or discouraging good practices was not mentioned.

Response: Sorry for the problems we have created. Hence we have included justification for favoring or discouraging good practices in the revised version of the manuscript.

#10. No comparison is made with studies from North African countries, Latin America countries and the like.

Response: thank you very much for your critical insights, hence we have tried our best to include literatures done in North African and Latin America. See the revised version of the manuscript.

#11. “This review revealed that the pooled estimated good practices of HCWM among healthcare workers in SSA were relatively low.” What does ‘relatively low mean?

Response: Sorry for the confusion we have created. Hence we have modified the way expression in the revised version to overcome this confusion. See the revised version of the manuscript.

#12. “The sex of the healthcare worker, the presence of waste management manuals/guidelines, knowledge of healthcare workers on healthcare waste management, training on waste management, and the working hours per day were factors significantly associated with HCWM practices among healthcare workers.“ Is it being male or female that tends to good practice? Does working less than or more than 8 hours favour good practice? Show the direction of each factor towards good practice.

Response: Thank you very much for your concern. All the reference variables are presented in brackets. The outcome of the study was expressed in terms of good practices. For example in this study female is presented as a reference and the odds is greater than one which implies that being male is odd times more likely to implement a good healthcare waste management practices than females. Similar principles will be applied for all other variables which have been declared as factors significantly associated with the outcome variables of the study.

Reviewer 2

#1. The pooled estimate of HCWM practices (50.42%) is based on studies with very high heterogeneity (I² = 97.7%). While subgroup analyses attempt to explore sources of heterogeneity, significant variability remains, limiting the generalizability of the results. Pooling data across diverse countries with varying healthcare systems and policies may mask important contextual factors influencing HCWM practices.

Response: thank you very much for your critical concern. Hence to find out the sources of heterogeneity we have performed meta-regression. The study setting is which is expected to share various aspects in terms of healthcare waste management practices. Therefore, for overcome such types of diversification, it is advisable to carry out Meta regression to determine the sources of heterogeneity. See the revised version of the manuscript.

#3. The exclusion of qualitative studies limits a deeper understanding of barriers and facilitators to HCWM practices, which is critical for policy and practice recommendations.

Response: Thank you very much for your critical concern. We have already excluded studies done on qualitative studies. This is because they have their methods of systematic review and meta-analysis. Therefore, it is difficult to manage both qualitative and quantitative studies simultaneously in a systematic review and meta-analysis. So we believe that systematic review for qualitative review should be done independently.

#4. Address the high heterogeneity more thoroughly by exploring meta-regression or other advanced statistical methods. Consider discussing the potential limitations of pooling data across diverse contexts in more depth. Appropriate statistical techniques were used, including random-effects modelling, Egger’s regression test for publication bias, and sensitivity analyses. Subgroup analyses by country, healthcare facility ownership, and pre- versus post-COVID-19 pandemic periods add depth to the findings. Despite subgroup analyses, high heterogeneity persists, which undermines the reliability of the pooled estimates. The justification for choosing the random-effects model is not explicitly stated, and alternative approaches (e.g., mixed-effects models) are not considered. The authors could have performed a meta-regression to explore potential moderators contributing to the observed heterogeneity. Provide a clearer rationale for model selection and explore advanced methods to account for heterogeneity. This would enhance the robustness of the findings.

Response: Thank you very much for your critical insight. Hence, we have conducted a Meta regression for find out the sources of heterogeneity. Additionally, we have included justification for selection of appropriate models in conducting systematic review and meta-analysis.

#5. The authors have stated that all relevant data are fully available within the manuscript and supporting information. References to included studies and the search strategy are provided, ensuring transparency and compliance with PLOS ONE’s data policy. The manuscript communicates the main findings and methodology effectively. There are several grammatical errors, typographical mistakes, and instances of unclear phrasing that impact the readability and professionalism of the manuscript. Examples include: Overly complex sentences, such as “The pooled estimate of good HCWM practices among study participants was found to be 50.42%...” These could be simplified for clarity.

Response: Sorry for the confusion we have created. We have amended all the issues you have raised. See the revised version of the manuscript.

#7. Repetitive phrases, such as “good healthcare waste management practices,” could be varied to improve flow.

Response: Thank you very much for your suggestion. Hence, we have tried to amend the problem in a revised version of the manuscript.

#8. Ambiguities in phrasing, such as “may cause the missing of some important studies,” should be revised to “may result in the omission of important studies

Response: Sorry for the confusion we have created. Hence, we have modified such types of unclear ideas throughout the manuscript. See the revised manuscript

#9. The manuscript requires editing for grammar, syntax, and readability to meet the standard of unambiguous English required by PLOS ONE. A professional language editing service may be beneficial.

Response: Thank you very much for your concern. Hence, we have amended the manuscript considering the PLOS ONE journal requirements and you suggestions. See the revised version of the manuscript.

#10. The manuscript outlines some strengths, such as the use of PRISMA guidelines, a comprehensive literature search, and robust inclusion criteria. However, the exclusion of non-English studies may introduce bias and limit the generalizability of findings, particularly in multilingual regions. The reliance on quantitative studies excludes valuable qualitative insights that could enrich understanding.

Response: Thank you very much for your concern. We have excluded studies done in languages other than English as a pre-determined criteria which is one of the strength of using a systematic review and meta-analysis. It is obviously clear than that we may miss pertinent findings done in another language. Because we do not have experts with other languages in understand their findings. Therefore, we have presented it as one of the limitations of this systematic review and meta-analysis. See the revised version of the manuscript.

#11. Expand the discussion of the study’s limitations, particularly regarding heterogeneity, language restrictions, and the exclusion of qualitative studies. This would provide a more balanced view of the study’s findings. No concerns regarding research ethics or dual publication were identified.

Response: thank you very much for your concern. Hence, we have included your suggestion in the revised version of the manuscript.

#12. The forest plots and PRISMA diagram are informative but could benefit from improved labelling to enhance clarity for readers.

Response: Thank you very much for your concern. Hence we have tried to present the figures based on the requirements of the journal using a PACE application for editing of graphs in PLOS ONE.

#13. While the authors provide recommendations for improving HCWM practices, these could be strengthened by incorporating actionable steps tailored to specific contexts (e.g., low-resource settings)

Response: Thank you very much for your concern. Hence, we have modified the recommendation as per your suggestion. See the revised version of the manuscript.

Reviewer 3

#1. Enhance the introduction by clearly articulating the novel contributions of the study, such as new insights into the factors influencing healthcare waste management practices in this specific regional context.

Response: Thank you very much for your critical concern. Hence, we have modified the introduction as per your recommendation. See the revised version of the manuscript.

#2. I recommend using abbreviations sparingly, only for frequently repeated terms, or providing an abbreviation guide with the full meanings.

Response: Sorry for the confusion we have created. Hence we have tried to use abbreviations and acronyms to lower the level. See the revised version of the manuscript.

#3. The reported high heterogeneity (I² = 97.7%) suggests substantial variability in the study outcomes, which could impact the generalizability of the findings. A more detailed exploration of the sources of this heterogeneity could strengthen the conclusions.

Response: Thank you very much for your concern. To overcome the sources of heterogeneity, we have performed meta-regression to enhance the generalizability of the findings. See the revised version of the manuscript.

#4. Some grammatical phrases need revision to enhance clarity and readability, e.g., "However, during the two independent reviewers, the issues were resolved through an evidence-based discussion

Response: Thank you very much for your concern. We have incorporated your comments in the revised version of the manuscript. See the revised version of the m

---

## [Decision Letter · Decision Letter 1]

18 Aug 2025

PONE-D-24-32196R1Healthcare waste management practice and associated factors among healthcare workers in Sub-Saharan Africa:  a systematic review and meta-analysis.PLOS ONE

Dear Dr. Berihun,

Thank you for submitting your manuscript to PLOS ONE. After careful consideration, we feel that it has merit but does not fully meet PLOS ONE’s publication criteria as it currently stands. Therefore, we invite you to submit a revised version of the manuscript that addresses the points raised during the review process. Dear author, try to address the comments given by the reviewers.

We look forward to receiving your revised manuscript.

Kind regards,

Tsegaye Alemayehu, Msc

Academic Editor

PLOS ONE

Journal Requirements:

Reviewers' comments:

Reviewer's Responses to Questions

**Comments to the Author**

1. If the authors have adequately addressed your comments raised in a previous round of review and you feel that this manuscript is now acceptable for publication, you may indicate that here to bypass the “Comments to the Author” section, enter your conflict of interest statement in the “Confidential to Editor” section, and submit your "Accept" recommendation.

Reviewer #1: All comments have been addressed

Reviewer #2: All comments have been addressed

Reviewer #3: (No Response)

2. Is the manuscript technically sound, and do the data support the conclusions?

Reviewer #1: Yes

Reviewer #2: Partly

Reviewer #3: Yes

3. Has the statistical analysis been performed appropriately and rigorously? 

Reviewer #1: Yes

Reviewer #2: Yes

Reviewer #3: Yes

4. Have the authors made all data underlying the findings in their manuscript fully available?

Reviewer #1: Yes

Reviewer #2: No

Reviewer #3: Yes

5. Is the manuscript presented in an intelligible fashion and written in standard English?

Reviewer #1: (No Response)

Reviewer #2: Yes

Reviewer #3: Yes

6. Review Comments to the Author

Reviewer #1: I congratulate authors for their significant efforts to improve the quality of revised manuscript. My comments are fully addressed and I found the revised manuscript acceptable for publication in the journal.

Reviewer #2: Thank you for your thoughtful revisions and efforts to improve the manuscript. Most of the previous concerns have been adequately addressed, particularly the improvements in methodology and interpretation. However, I encourage you to provide more detailed reporting of the meta-regression results (e.g., coefficients, p-values), and ensure that the dataset used for the analysis is made publicly available in line with PLOS ONE’s data policy. These are minor revisions and can be addressed without extensive changes. So I recommend minor revisions to address some remaining concerns prior to publication.

While the revised manuscript represents a significant improvement in clarity, structure, and methodological explanation, several technical concerns remain that limit the overall robustness and reproducibility of the findings:

1. The meta-analysis continues to exhibit very high heterogeneity (I² = 97.7%), which is only partially addressed through subgroup analyses and meta-regression. Although the authors explored variability by country, facility ownership, and the COVID-19 period, residual heterogeneity remains substantial. This weakens the strength of the pooled estimate and limits the generalizability of the findings. Moreover, the inclusion of studies from diverse healthcare systems with varying definitions of “good HCWM practices” introduces further contextual variability that is not fully accounted for in the interpretation. The decision to exclude non-English and qualitative studies—while methodologically valid—reduces the comprehensiveness of the evidence base. Policy recommendations, though revised, are still somewhat generic and not consistently anchored in the patterns identified through analysis.

The authors should elaborate on the meta-regression findings by reporting key statistics such as p-values, regression coefficients, and the proportion of variance explained (R²). Additionally, they should explicitly discuss how differences in healthcare systems and definitions of “good HCWM practices” across countries may limit the generalizability of the pooled estimate.

2. The authors have strengthened the statistical methods in the revised version by applying random-effects modeling, performing subgroup analyses and meta-regression, and assessing publication bias and sensitivity. However, the rationale for selecting specific models, such as the random-effects approach over mixed-effects or Bayesian models, could be more thoroughly articulated. Importantly, the results of the meta-regression are not adequately detailed—regression coefficients, p-values, and explained variance (e.g., R²) are not reported, limiting the reader’s ability to assess its effectiveness. Confidence intervals for subgroup estimates and heterogeneity statistics within subgroups should also be reported and discussed more explicitly. Finally, assumptions underlying the statistical models are not addressed.

While random-effects models are appropriate, the rationale for their selection over alternative models (e.g., mixed-effects or Bayesian approaches) should be explained more clearly. The manuscript should also provide more detailed reporting of subgroup analyses, including subgroup-level heterogeneity (I²) and confidence intervals for each pooled estimate.

3. The current Data Availability Statement does not fully comply with PLOS ONE’s requirements. While the authors state that all relevant data are within the manuscript and Supporting Information, no raw data (e.g., extracted prevalence rates, sample sizes, standard errors) are provided. There is no supplementary data table summarizing the inputs used for meta-analysis, nor is there a public repository link or access to analysis code. This lack of raw data and transparency hinders reproducibility and limits the scientific value of the review.

The authors should include a detailed table listing study-level data extracted for the meta-analysis (e.g., sample sizes, effect sizes, standard errors), and ideally upload the dataset to a public repository (e.g., OSF, Figshare). They should also consider sharing the statistical code used for analysis to enhance transparency and reproducibility in line with PLOS ONE’s data policy.

In summary, while the manuscript now presents a clearer and more technically sound review, further improvements are needed in the justification and reporting of statistical methods and in meeting open data standards to ensure full transparency and reliability of the findings.

Reviewer #3: The authors have addressed many reviewer concerns and made commendable progress in revising the manuscript; however, several methodological, reporting, and clarity issues persist, particularly surrounding heterogeneity handling, framing of pooled estimates, language quality, and overgeneralization based on data from a few countries.

The high heterogeneity remains a serious limitation to the pooled estimate’s reliability. While a meta-regression was conducted and a table provided the results, the interpretation is not sufficiently detailed in the Results or Discussion. Include a clear interpretation of key moderators.

Over 60% of the included studies are from Ethiopia. While SSA is in the title and rationale, this regional label gives a false sense of geographic representativeness. Either adjust the title to reflect “Eastern Africa” or reframe the findings throughout to highlight this imbalance

Authors state “≥50% of recommended practices” as the threshold, but this criterion is inconsistent across studies.

Although improvements were made, multiple instances of awkward phrasing, grammatical errors, and redundant repetition persist.

7. PLOS authors have the option to publish the peer review history of their article (what does this mean? ). If published, this will include your full peer review and any attached files.

**Do you want your identity to be public for this peer review?** For information about this choice, including consent withdrawal, please see our Privacy Policy .

Reviewer #1: **Yes: ** Taffere Addis, Addis Ababa University, Addis Ababa, Ethiopia

Reviewer #2: **Yes: ** Volkan Medeni

Reviewer #3: No

---

## [Author Response · Author response to Decision Letter 2]

9 Sep 2025

Rebuttal letter

Response to the editor and reviewers.

Reviewer 1: has no concerns.

Response to Reviewer 2

#1. The meta-analysis continues to exhibit very high heterogeneity (I² = 97.7%), which is only partially addressed through subgroup analyses and meta-regression. Although the authors explored variability by country, facility ownership, and the COVID-19 period, residual heterogeneity remains substantial. This weakens the strength of the pooled estimate and limits the generalizability of the findings. Moreover, the inclusion of studies from diverse healthcare systems with varying definitions of “good HCWM practices” introduces further contextual variability that is not fully accounted for in the interpretation. The decision to exclude non-English and qualitative studies—while methodologically valid—reduces the comprehensiveness of the evidence base. Policy recommendations, though revised, are still somewhat generic and not consistently anchored in the patterns identified through analysis.

The authors should elaborate on the meta-regression findings by reporting key statistics such as p-values, regression coefficients, and the proportion of variance explained (R²). Additionally, they should explicitly discuss how differences in healthcare systems and definitions of “good HCWM practices” across countries may limit the generalizability of the pooled estimate.

Response: Thank you sincerely for your valuable and constructive feedback aimed at improving the quality of our manuscript. As highlighted in the results section, the heterogeneity among the included studies was notably high, with an I² value approaching 97%, indicating substantial variability. To explore potential sources of this heterogeneity, we conducted subgroup analyses; however, none of the examined variables accounted for the observed variation. Consequently, we proceeded with a meta-regression analysis to further investigate the origins of heterogeneity. As presented in the meta-regression table, all variables exhibited confidence intervals that spanned both negative and positive values, suggesting no significant contribution to heterogeneity. This interpretation is further supported by the p-values, all of which exceeded the conventional threshold of statistical significance (p > 0.05), reinforcing the conclusion that the variables assessed were not responsible for the heterogeneity. Regarding outcome measurement, although the included studies employed various assessment tools, the majority defined "good practice" as achieving a score of 50% or higher on the recommended preventive measures. Accordingly, this has been acknowledged as one of the limitations of the present systematic review and meta-analysis

#2. The authors have strengthened the statistical methods in the revised version by applying random-effects modelling, performing subgroup analyses and meta-regression, and assessing publication bias and sensitivity. However, the rationale for selecting specific models, such as the random-effects approach over mixed-effects or Bayesian models, could be more thoroughly articulated. Importantly, the results of the meta-regression are not adequately detailed—regression coefficients, p-values, and explained variance (e.g., R²) are not reported, limiting the reader’s ability to assess its effectiveness. Confidence intervals for subgroup estimates and heterogeneity statistics within subgroups should also be reported and discussed more explicitly. Finally, assumptions underlying the statistical models are not addressed.

While random-effects models are appropriate, the rationale for their selection over alternative models (e.g., mixed-effects or Bayesian approaches) should be explained more clearly. The manuscript should also provide more detailed reporting of subgroup analyses, including subgroup-level heterogeneity (I²) and confidence intervals for each pooled estimate.

Response: The fixed effect model and random effects model are the two most commonly used approaches in systematic reviews and meta-analyses. Since this systematic review and meta-analysis focus on Sub-Saharan African countries, it is anticipated that there will be variability in the factors contributing to the outcomes. The fixed effect model is appropriate when the study populations are relatively homogeneous regarding these contributing factors. However, our research encompasses a vast geographic region that is likely to exhibit substantial heterogeneity. For this reason, the random effects model is recommended to assess the degree of heterogeneity across studies. When heterogeneity is minimal, typically less than 25%, the fixed effect model is preferable. Conversely, if heterogeneity is substantial, a random effects model is more suitable. In our analysis, the level of heterogeneity is very high, close to 97%, which justifies the use of the random effects model in our study.

#3. The current Data Availability Statement does not fully comply with PLOS ONE’s requirements. While the authors state that all relevant data are within the manuscript and Supporting Information, no raw data (e.g., extracted prevalence rates, sample sizes, standard errors) are provided. There is no supplementary data table summarising the inputs used for meta-analysis, nor is there a public repository link or access to analysis code. This lack of raw data and transparency hinders reproducibility and limits the scientific value of the review.

The authors should include a detailed table listing study-level data extracted for the meta-analysis (e.g., sample sizes, effect sizes, standard errors), and ideally upload the dataset to a public repository (e.g., OSF, Figshare). They should also consider sharing the statistical code used for analysis to enhance transparency and reproducibility in line with PLOS ONE’s data policy.

In summary, while the manuscript now presents a clearer and more technically sound review, further improvements are needed in the justification and reporting of statistical methods and in meeting open data standards to ensure full transparency and reliability of the findings.

Response: Thank you sincerely for your valuable and thoughtful feedback. In response to your concerns, we have supplied all the necessary supplementary materials required to support the comprehensive analysis presented in this systematic review and meta-analysis. These additional details ensure the transparency and completeness of our work, facilitating a deeper understanding and validation of our findings.

Review 3:

#1. The authors have addressed many reviewer concerns and made commendable progress in revising the manuscript; however, several methodological, reporting, and clarity issues persist, particularly surrounding heterogeneity handling, framing of pooled estimates, language quality, and overgeneralization based on data from a few countries. The high heterogeneity remains a serious limitation to the pooled estimate’s reliability. While a meta-regression was conducted and a table provided the results, the interpretation is not sufficiently detailed in the Results or Discussion. Include a clear interpretation of key moderators. Over 60% of the included studies are from Ethiopia. While SSA is in the title and rationale, this regional label gives a false sense of geographic representativeness. Either adjust the title to reflect “Eastern Africa” or reframe the findings throughout to highlight this imbalance. Authors state “≥50% of recommended practices” as the threshold, but this criterion is inconsistent across studies. Although improvements were made, multiple instances of awkward phrasing, grammatical errors, and repetition persist.

Response: Thank you very much for your valuable and constructive feedback, which has helped improve the quality of our manuscript. We have employed several strategies to address heterogeneity among the included studies. These include statistical tests for heterogeneity such as Cochran’s Q test and the I² statistic, and the use of an appropriate meta-analytic approach—a random effects model—due to heterogeneity levels exceeding 50%. Additionally, we explored sources of heterogeneity through subgroup analyses, meta-regression, and sensitivity analyses.

Regarding the framing of the pooled estimate, this systematic review and meta-analysis were conducted following standard guidelines, including clearly defining the research question, data extraction, and quality assessment protocols. Concerning language quality, we have carefully revised the entire manuscript to address language issues and improve clarity.

On the topic of overgeneralization, it should be noted that inclusion criteria were pre-established to minimize bias. While Sub-Saharan Africa comprises nearly 49 countries, our review includes studies from only nine countries across different regions of Africa, as determined by our protocol before the literature search. Therefore, we could not alter the study scope afterwards. Most included studies are from East Africa, particularly Ethiopia, but we also incorporated research from Western and Southern Africa. Based on this, we believe changing the title to focus solely on East Africa is not appropriate; however, we acknowledge this as a limitation in our review. The issue of high heterogeneity remains, with an I² of 96.8%, which is a significant concern in the manuscript through detailed explanation in the results and discussion sections. Regarding outcome measurement, although there were variations in measurement tools, most studies classified good practices as scores of 50% or above. Finally, we have made extensive efforts to correct phrasing, grammatical errors, and remove redundant ideas throughout the revised manuscript to enhance readability and quality.

---

## [Decision Letter · Decision Letter 2]

24 Sep 2025

Healthcare waste management practice and associated factors among healthcare workers in Sub-Saharan Africa:  a systematic review and meta-analysis.

PONE-D-24-32196R2

Dear Dr. Berihun,

We’re pleased to inform you that your manuscript has been judged scientifically suitable for publication and will be formally accepted for publication once it meets all outstanding technical requirements.

Kind regards,

Tsegaye Alemayehu, Msc

Academic Editor

PLOS ONE

Additional Editor Comments (optional):

Reviewer #1:

Reviewer #2:

Reviewers' comments:

Reviewer's Responses to Questions

**Comments to the Author**

1. If the authors have adequately addressed your comments raised in a previous round of review and you feel that this manuscript is now acceptable for publication, you may indicate that here to bypass the “Comments to the Author” section, enter your conflict of interest statement in the “Confidential to Editor” section, and submit your "Accept" recommendation.

Reviewer #1: All comments have been addressed

Reviewer #2: All comments have been addressed

2. Is the manuscript technically sound, and do the data support the conclusions?

Reviewer #1: Yes

Reviewer #2: Yes

3. Has the statistical analysis been performed appropriately and rigorously? 

Reviewer #1: Yes

Reviewer #2: Yes

4. Have the authors made all data underlying the findings in their manuscript fully available?

Reviewer #1: Yes

Reviewer #2: Yes

5. Is the manuscript presented in an intelligible fashion and written in standard English?

Reviewer #1: Yes

Reviewer #2: Yes

6. Review Comments to the Author

Reviewer #1: The second revised version is significantly improved and publishable in its current form. The methodologies followed, the statistical analysis, results and interpretation are valid and appropriate. It is highly informative for decision makers, managers and practitioners in the area of healthcare waste management.

Reviewer #2: I would like to thank the authors for their thoughtful and detailed responses to the reviewers' comments, and for submitting a substantially improved version of the manuscript. I appreciate the effort put into clarifying the methodology, expanding the discussion, and refining the manuscript’s structure and language. The concerns I previously raised have been sufficiently addressed. I recommend accepting the manuscript in its current form or with minor editorial revisions.

7. PLOS authors have the option to publish the peer review history of their article (what does this mean? ). If published, this will include your full peer review and any attached files.

**Do you want your identity to be public for this peer review?** For information about this choice, including consent withdrawal, please see our Privacy Policy .

Reviewer #1: **Yes: ** Taffere Addis, Addis Ababa University, Ethiopia.

Reviewer #2: No

---

## [Editor Report · Acceptance letter]

PONE-D-24-32196R2

PLOS ONE

Dear Dr. Berihun,

I'm pleased to inform you that your manuscript has been deemed suitable for publication in PLOS ONE. Congratulations! Your manuscript is now being handed over to our production team.

Kind regards,

on behalf of

Dr. Tsegaye Alemayehu

Academic Editor

PLOS ONE